# GRAPH POSITIONAL AND STRUCTURAL ENCODER

## ABSTRACT

Positional and structural encodings (PSE) enable better identifiability of nodes within a graph, as in general graphs lack a canonical node ordering. This renders PSEs essential tools for empowering modern GNNs, and in particular graph Transformers. However, designing PSEs that work optimally for a variety of graph prediction tasks is a challenging and unsolved problem. Here, we present the graph positional and structural encoder (GPSE), a first-ever attempt to train a graph encoder that captures rich PSE representations for augmenting any GNN. GPSE can effectively learn a common latent representation for multiple PSEs, and is highly transferable. The encoder trained on a particular graph dataset can be used effectively on datasets drawn from significantly different distributions and even modalities. We show that across a wide range of benchmarks, GPSE-enhanced models can significantly improve the performance in certain tasks, while performing on par with those that employ explicitly computed PSEs in other cases. Our results pave the way for the development of large pre-trained models for extracting graph positional and structural information and highlight their potential as a viable alternative to explicitly computed PSEs as well as to existing self-supervised pre-training approaches.

## 1 INTRODUCTION

Graph neural networks (GNN) are the dominant paradigm in graph representation learning (Hamilton et al., 2017b; Bronstein et al., 2021), spanning diverse applications across many domains in biomedicine (Yi et al., 2022), molecular chemistry (Xia et al., 2022), and more (Dwivedi et al., 2022a; Hu et al., 2020a; 2021; Liu et al., 2022a). For most of its relatively short history, GNN algorithms were developed within the message-passing neural network (MPNN) framework (Gilmer et al., 2017), where vertices exchange internal states within their neighborhoods defined by the graph structure, which are typically sparse. Despite being computationally efficient, the sparsity leveraged by MPNN has raised many fundamental limits, such as the 1-WL bounded expressiveness (Xu et al., 2019), under-reaching (Barceló et al., 2020), and over-squashing (Alon and Yahav, 2021; Topping et al., 2022). More recently, leveraging the success of the Transformer model in natural language processing (Vaswani et al., 2017), graph Transformer (GT) models were developed as a new paradigm for GNN to address the above limitations (Dwivedi and Bresson, 2021). Attending to all pairs of nodes in a graph circumvents the aforementioned sparsity-induced limitations of MPNN, but it also discards all inductive biases relating to the graph structure (Battaglia et al., 2018), which MPNNs leverage well. Thus, reintroducing such inductive bias via positional and structural encodings (PSE) has been one of the most essential steps that led to the early success of GT (Rampášek et al., 2022; Dwivedi and Bresson, 2021; Ying et al., 2021).

While many different types of PSEs have been hand-crafted and used by various GT models, a consensus on the best PSE to use in a general context has yet to be reached. In fact, the optimal choice of PSE often depends on the specific task at hand. For example, random walk encodings are typically effective for small molecular tasks (Rampášek et al., 2022; Dwivedi et al., 2022b), while graph Laplacian eigenvectors might be more suitable for tasks that involve long-range dependencies (Dwivedi et al., 2022c;a). Therefore, developing a systematic approach to learn a unified encoding that integrates various PSEs remains an open challenge.

Encoding information that is transferable between certain graph datasets and tasks is typically achieved via self-supervised learning (SSL) methods. Pre-training GNNs via SSL has proven effective in learning graph features in data-abundant settings, which can then be transferred to downstream tasks (Hu et al., 2020b; Wang et al., 2022a; Xie et al., 2022). However, SSL on graphs

is plagued by a critical drawback: SSL that performs well on one downstream task may not help or even lead to negative transfer on another, and the success of graph SSL methods typically hinges on whether the data for the pre-training and downstream tasks are well-aligned (Sun et al., 2022; Hu et al., 2020b; Wang et al., 2022a).

**Contributions** In this work, we approach the abovementioned problems by introducing the graph positional and structural encoder (GPSE). We summarize our main contributions as follows.

1. We propose GPSE, the first attempt at training a graph encoder that extracts rich *positional and structural representations* from graph structures, which can be applied to any MPNN or GT model as a replacement for explicitly constructed PSEs.
2. We show the superior performance of GPSE over traditionally used PSEs across a variety of benchmarks, even achieving SOTA performance on ZINC and Peptides-struct datasets.
3. Extensive experiments demonstrate that GPSE is highly *transferable* across graphs of different sizes, connectivity patterns, and modalities.

## 1.1 RELATED WORK

**Positional and structural encodings (PSE)** *Positional encoding* was originally implemented as a series of sinusoidal functions in the Transformer model to capture the ordinal position of words in a sentence (Vaswani et al., 2017). However, capturing the position of nodes in a graph is harder, as in general, such a canonical ordering does not exist in graphs. Many recent works on graph Transformers (GT) use the graph Laplacian eigenvectors as the positional encodings (Rampášek et al., 2022; Kreuzer et al., 2021), which are direct analogues to the sinusoids in Euclidean space (Spielman, 2012). Other methods for encoding positional information include electrostatic potential encodings (Kreuzer et al., 2021), shortest-path distances (Ying et al., 2021), tree-based encodings (Shiv and Quirk, 2019), etc. *Structural encodings*, on the other hand, have been developed particularly on graph-structured data to encode rich local and global connectivity patterns. The random walk encoding, for example, has been proven to be powerful enough to distinguish $r$-regular graphs with high probabilities (Li et al., 2020; Dwivedi et al., 2022b). It has also shown great performance when used with GT models, particularly on small molecular graph benchmarks (Rampášek et al., 2022; Dwivedi and Bresson, 2021; Dwivedi et al., 2022b). Other notable structural encodings include the heat kernel (Kreuzer et al., 2021; Mialon et al., 2021), subgraph counting (Bouritsas et al., 2022; Zhao et al., 2022), node degree centralities (Ying et al., 2021), etc. The usage of PSEs has also demonstrated success in MPNN, besides GT, as additional node features that are combined with the original graph features (Dwivedi et al., 2022a; Lim et al., 2022; Dwivedi et al., 2022b; Wang et al., 2022b; Dwivedi et al., 2022b). Despite this great amount of work in developing different PSEs and ways to extract invariants from them further (Lim et al., 2022; Chen et al., 2022), it is still unclear how to systematically encode information from multiple types of PSEs to effectively augment GNNs.

**Self-supervised learning (SSL)** The general goal of SSL is learning to extract useful graph representations from large amounts of unlabeled, but typically featured, graphs (Xie et al., 2022; Liu et al., 2022b). One popular approach to achieve this is *contrastive learning*, which aims to embed corrupted or augmented views of the same graph into more similar representations than those from other graphs. Most contrastive graph pre-training methods generally differ in (i) what type of graphs and associated domain-specific information they use, such as 3D and geometric molecular structure (Stärk et al., 2022; Li et al., 2022), (ii) how they generate contrastive views (Hu et al., 2020b; You et al., 2020b), and (iii) what contrastive loss they use (Veličković et al., 2019; You et al., 2020b; Jiao et al., 2020). *Predictive learning*, on the other hand, aims to recover graph properties that are (a) intrinsic, such as the adjacency matrix (Kipf and Welling, 2016; Grover et al., 2019) , or (b) extrinsic, such as molecular motif predictions (Rong et al., 2020a). However, the above-mentioned works have limited transferability as they all rely on domain-specific graph features or self-supervision. A fully generalizable and transferable SSL graph encoding method that can be trained on *unfeatured* and *unlabeled* graph datasets is currently lacking.

## 2 METHODS

Our core idea is to train an MPNN as the graph encoder to extract rich positional and structural representations of any query graph based *solely* on its graph structure (Figure 1A). To achieve this,

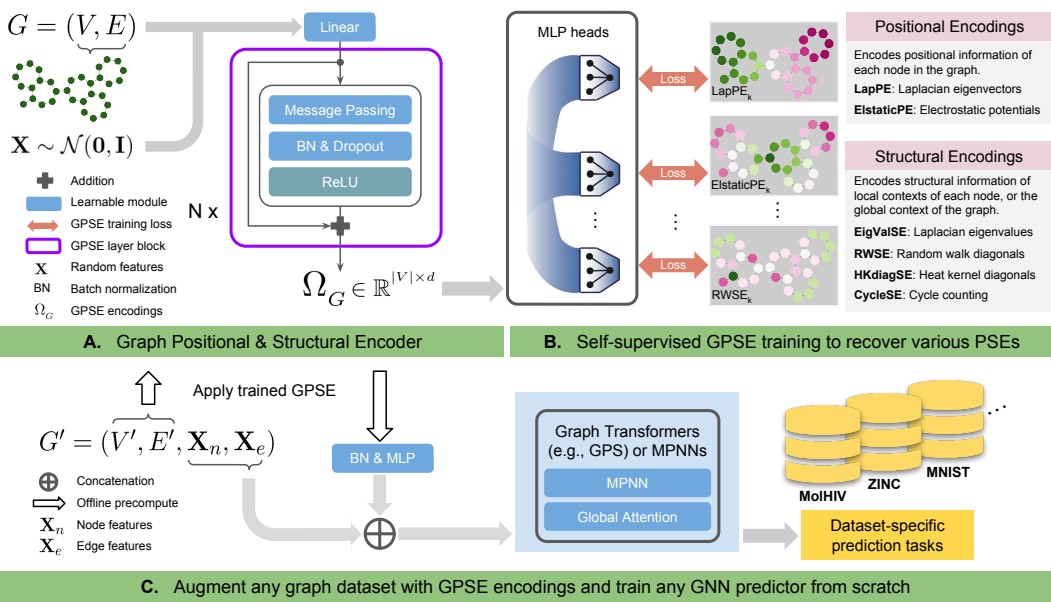

Figure 1: Overview of Graph Positional and Structural Encoder (GPSE) training and application.

we design a collection of PSEs encompassing a broad range of encodings and use them as self-supervision to train the encoder (Figure 1B). Once the encoder is trained on graphs from a particular dataset, it can extract PSE representations for any downstream dataset as the encodings to augment any downstream model (Figure 1C).

For downstream tasks, we primarily build on top of a powerful graph Transformer framework, GPS (Rampášek et al., 2022), that leverages the advantages of both the inductive bias of the local message passing (Battaglia et al., 2018) and the expressiveness of the global attention (Vaswani et al., 2017). We also demonstrate GPSE's usefulness for more general MPNN in our experiments.

## 2.1 SELF-SUPERVISION VIA POSITIONAL AND STRUCTURAL ENCODINGS (PSE)

We design a diverse collection of six PSEs for GPSE to learn against, including the Laplacian eigenvectors (4) and eigenvalues (4), the electrostatic positional encodings (7), the random walk structural encodings (20), the heat kernel structural encodings (20), and the cycle counting graph encodings (7). In short, positional encodings informs the relative position of each node in the graph, while structural encodings describe the local connectivity patterns around a node (Figure 1). See Appendix A for their precise mathematical definitions.

## 2.2 GPSE ARCHITECTURE

Over-smoothing and over-squashing are two key challenges to overcome when learning with deep MPNNs. Thus, our GPSE model also has to overcome these challenges to be able to learn good joint representations for PSEs. In this section, we present different architectural choices and their relationship with these two phenomena, as well as theoretical justifications behind our choices. Please refer to Appendix C for technical definitions; we also discuss the relevance of over-smoothing and over-squashing with GPSE in further detail in Appendix C.1.

Evidence suggests that over-smoothing and squashing in graph networks relate to the graph's curvature (Topping et al., 2022; Nguyen et al., 2022; Giraldo et al., 2022). There are many versions of graph curvature (Forman, 2003; Ollivier, 2009; Sreejith et al., 2016; Topping et al., 2022) but intuitively, they encode how a node's neighborhood looks like a clique (positive curvature), a grid (zero curvature), or a tree (negative curvature). A clique's high connectivity leads to rapid smoothing, while a tree's exponentially increasing size of $k$-hop neighborhood causes over-squashing. These phenomena are in competition, and negating both is impossible in an MPNN using graph rewiring (architectures using modified adjacency for node aggregation). However, there seems to be a *sweet*

*spot* where both effects are not minimized but the sum of the two effects is minimized. This minimum is sought in Giraldo et al. (2022) and Nguyen et al. (2022). Some of our following choices of architecture are justified by this search of a sweet spot in the smoothing-squashing trade-off.

**Deep GNN**  As several of the target PSEs, such as the Laplacian eigenvectors, require having a global view of the query graph, it is crucial for the encoder to capture long-range dependencies accurately. To accomplish this, we need to use an unconventionally deep MPNN with 20 layers. However, if a graph network suffers from over smoothing, having this many layers will result in approximately uniform node features (Oono and Suzuki, 2020; Li et al., 2019).

**Residual connection**  A first attempt at reducing the smoothing is to exploit the proven ability of residual connections in reducing over smoothing (Li et al., 2019).

**Gating mechanism**  Using gating mechanism in the aggregation helps reduce the over smoothing even further. Indeed, gating allows the network to reduce the weight of some edges and in the limit effectively re-wire the graph by completely or partially ignoring some edges. Essentially, we argue that it is possible for gating to act as a graph sparsification device, which decreases the graph curvature and have been shown to alleviate over-smoothing (Giraldo et al., 2022; Rong et al., 2020b).

**Virtual node**  In addition, we use virtual node (VN) (Gilmer et al., 2017) to enable global message passing; as the virtual node has access to the states of all nodes, it allows for (a) better representation of graph-level information and (b) faster propagation of information between nodes that are further apart, and thus faster convergence of states. In technical terms, adding the virtual node drastically increases the connectivity of the graph and in turn its curvature (Appendix C, prop. 1), and consequently decreases the over squashing. Alternatively, one can see that the Cheeger constant (another measure of *bottleneckness* (Topping et al., 2022)) of the graph increases after adding the virtual node.

**Random node features**  One critical question is whether an MPNN is expressive enough to learn all the target PSEs. In particular, some PSEs, such as the Laplacian eigenvalues, may require distinguishability beyond 1-WL (Fürer, 2010). Despite the known 1-WL expressiveness limitation of a standard MPNN when using constant node features (Xu et al., 2019), many recent works have shown that random node features can help MPNNs surpass the 1-WL expressiveness(Sato et al., 2021; Abboud et al., 2021; Kanatsoulis and Ribeiro, 2022). Thus, we base our encoder architecture on an MPNN coupled with random input node features, as shown in Figure 1A.

We argue and later validate that together, the above architectural design choices lead to an effective graph encoder that finds the balance between smoothing and squashing (§3.4), and even has an elevated expressiveness due to the random features (§3.3). A detailed ablation study to highlight the impoartance of our architectural choices is also available in Table F.1.

## 2.3  TRAINING GPSE

Given a query graph structure $G = (V, E)$, we first generate a $k$-dimensional feature $\mathbf{X} \sim \mathcal{N}(\mathbf{0}, \mathbf{I})$ from a standard normal distribution for each node, which is then passed through a linear projection layer to match the $d$-dimensional hidden dimension. Then, the projected features and the graph structure are processed by $N$ message-passing layers with residual connections (Li et al., 2019), resulting in the final GPSE representations Figure 1. To train the encoder using the target PSEs, we decode the GPSE using multiple independent MLP heads, one per PSE, and compute the loss based on the sum of $\ell_1$ and cosine similarity losses. This learning approach falls into the category of predictive learning. We note that contrastive approaches are infeasible for learning PSE representations as it is undesirable to obtain a representation that is insensitive to structural perturbations.

**Training dataset**  PCQM4Mv2 (Hu et al., 2021) is a typical choice of pre-training dataset for molecular tasks. However, since GPSE only extracts features from graph structures (e.g., methane, CH4, would be treated as the same graph as silane, SiH4), the amount of training samples reduces to 273,920 after extracting unique graphs. Instead, we train GPSE with MolPCBA (Hu et al., 2020a), which contains 323,555 unique molecular graphs, with an average number of 25 nodes. We randomly select 5% validation and 5% testing data fixed across runs, and use the remaining data for training.

## 3  EXPERIMENTS

**GPSE successfully predicts a wide range of target PSEs**  The self-supervised goal of our GPSE is to learn graph representations from which it is possible to recover predefined positional and structural encodings. For each PSE type, we quantify the prediction performance in terms of the coefficient of determination ($R^2$) scores, as presented in Table 1. When trained on a 5% (16,177) subset of MolPCBA molecular graphs, our GPSE achieves 0.9790 average test $R^2$ score across the 6 PSEs. Further, we show that the test performance improves asymptotically as the number of training samples increases (§3.4), achieving 0.9979 test $R^2$ when training on 90% (291,199) of MolPCBA unique graphs. These results demonstrate the ability of the GPSE to extract rich positional and structural information from a query graph, as well as its ability to learn from an increasing amount of data.

Table 1: Held-out PSE prediction performance of GPSE on 5% MolPCBA.

| PSE | $R^2 \uparrow$ |
|---|---|
| ElstaticPE | 0.964 |
| LapPE | 0.973 |
| RWSE | 0.984 |
| HKdiagSE | 0.981 |
| EigValSE | 0.982 |
| CycleSE | 0.977 |
| Overall | 0.979 |

### 3.1 Enhancing performance on molecular graph datasets

In this series of experiments, we demonstrate that GPSE is a general augmentation to a wide range of GNN models and is a viable alternative to existing SSL pre-training approaches.

**GPSE-augmented GPS is highly competitive on molecular graph benchmarks**  In this set of experiments, we compare performance of the GPS model augmented with our GPSE encodings versus the same model using (a) no PSE, (b) random features as PSE, and (c) LapPE and RWSE (two PSEs from §2.1) on four common molecular property prediction benchmarks (Dwivedi et al., 2022a; Hu et al., 2020a; 2021). For ZINC (Gómez-Bombarelli et al., 2018) and PCQM4Mv2 (Hu et al., 2020a), we use their subset versions following Dwivedi et al. (2022a) and Rampášek et al. (2022), respectively.

Table 2: Performance in four molecular property prediction tasks, averaged over 10 seeds.

| | **ZINC** (subset) **MAE** $\downarrow$ | **PCQM4Mv2** (subset) **MAE** $\downarrow$ | **MolHIV** **AUROC** $\uparrow$ | **MolPCBA** **AP** $\uparrow$ |
|---|---|---|---|---|
| GCN (Kipf and Welling, 2017) | 0.3670 ± 0.0110 | – | 0.7599 ± 0.0119 | 0.2424 ± 0.0034 |
| GIN (Xu et al., 2019) | 0.5260 ± 0.0510 | – | 0.7707 ± 0.0149 | 0.2703 ± 0.0023 |
| CIN (Bodnar et al., 2021) | 0.0790 ± 0.0060 | – | **0.8094 ± 0.0057** | – |
| CRaWI (Toenshoff et al., 2021) | 0.0850 ± 0.0040 | – | – | **0.2986 ± 0.0025** |
| GPS+rand | 0.8766 ± 0.0107 | 0.4768 ± 0.0171 | 0.6210 ± 0.0444 | 0.0753 ± 0.0045 |
| GPS+none | 0.1182 ± 0.0049 | 0.1329 ± 0.0030 | 0.7798 ± 0.0077 | 0.2869 ± 0.0012 |
| GPS+LapPE | 0.1078 ± 0.0084 | 0.1267 ± 0.0004 | 0.7736 ± 0.0097 | 0.2939 ± 0.0016 |
| GPS+RWSE | 0.0700 ± 0.0040 | 0.1230 ± 0.0008 | 0.7880 ± 0.0101 | 0.2907 ± 0.0028 |
| GPS+AllPSE | 0.0734 ± 0.0030 | 0.1254 ± 0.0011 | 0.7645 ± 0.0236 | 0.2826 ± 0.0001 |
| GPS+GPSE | **0.0648 ± 0.0030** | **0.1196 ± 0.0004** | 0.7815 ± 0.0133 | 0.2911 ± 0.0036 |

We first highlight that, with GPSE-augmented input features, GPS achieves a remarkable 0.0648 MAE on ZINC (Table 2). Moreover, GPSE always improves the baseline by a margin similar to or higher than standard PSEs. By improving on GPS+RWSE, GPS+GPSE also becomes the best model on PCQM4Mv2 that (a) is not an ensemble method and (b) does not have access to 3D information.

**GPSE as a universal PSE augmentation**  The utility of GPSE encodings is not specific to GPS. On ZINC (12k subset), we show that augmenting different MPNN methods and the Transformer with GPSE universally results in significant improvements: 56.24% reduction in test MAE on average compared to baselines that do not make use of any PSE (Table 3). We perform the same set of experiments on PCQM4Mv2 as well (Table F.2), and obtain similar improvements on explicitly computed PSEs that validate the success of GPSE.

**Feature augmentation using GPSE vs. SSL pre-training**  Our GPSE feature augmentation approach is related to the SSL pre-training approaches (Hu et al., 2020b; You et al., 2020b; Xie et al., 2022; Xu et al., 2021) in that both transfer knowledge from a large pre-training dataset to another for downstream evaluation. Yet, at the same time, our approach is a notable departure from previous SSL approaches in two distinct aspects:

1. The trained GPSE model is only used as a feature extractor that can be coupled with *any* type of downstream prediction model, which will be trained from scratch.

Table 3: Four PSE augmentations combined with five different GNN models evaluated on ZINC (12k subset) dataset. Performance is evaluated as MAE (↓) and averaged over 4 seeds.

| | GCN | GatedGCN | GIN | GINE | Transformer | Avg. reduction |
|---|---|---|---|---|---|---|
| none | 0.288 ± 0.004 | 0.236 ± 0.008 | 0.285 ± 0.004 | 0.118 ± 0.005 | 0.686 ± 0.017 | – |
| rand | 1.277 ± 0.340 | 1.228 ± 0.012 | 1.239 ± 0.011 | 0.877 ± 0.011 | 1.451 ± 0.002 | N/A |
| LapPE | 0.209 ± 0.008 | 0.194 ± 0.006 | 0.214 ± 0.004 | 0.108 ± 0.008 | 0.501 ± 0.145 | 21.12% |
| RWSE | 0.181 ± 0.003 | 0.167 ± 0.002 | 0.175 ± 0.003 | 0.070 ± 0.004 | 0.219 ± 0.007 | 42.75% |
| AllPSE | 0.150 ± 0.007 | 0.143 ± 0.007 | 0.153 ± 0.006 | 0.073 ± 0.003 | 0.190 ± 0.008 | 50.85% |
| GPSE | **0.129 ± 0.003** | **0.113 ± 0.003** | **0.124 ± 0.002** | **0.065 ± 0.003** | **0.189 ± 0.016** | **56.24%** |

2. GPSE extracts representations *solely* from the graph structure and does not make use of the domain-specific features such as atom and bond types (Hu et al., 2020b), allowing GPSE to be utilized on general graph datasets.

To compare the performance of SSL pre-training and GPSE feature augmentation, we use the MoleculeNet (Wu et al., 2018; Hu et al., 2020b) datasets. For the downstream model, we use the identical GINE architecture Hu et al. (2020b) from Sun et al. (2022). Finally, the extracted features from the GPSE are concatenated with the initial atom embeddings and are then fed into the GINE model.

Table 4: Performance on MoleculeNet small datasets (scaffold test split), evaluated in AUROC (%) ↑. Red indicates worse than baseline performance.

| | BBBP | BACE | Tox21 | ToxCast | SIDER | ClinTox | MUV | HIV |
|---|---|---|---|---|---|---|---|---|
| No pre-training (baseline) (Hu et al., 2020b) | 65.8 ± 4.5 | 70.1 ± 5.4 | 74.0 ± 0.8 | 63.4 ± 0.6 | 57.3 ± 1.6 | 58.0 ± 4.4 | 71.8 ± 2.5 | 75.3 ± 1.9 |
| Self-supervised pre-trained (Hu et al., 2020b) | 68.8 ± 0.8 | 79.9 ± 0.9 | 76.7 ± 0.4 | 64.2 ± 0.5 | 61.0 ± 0.7 | 71.8 ± 4.1 | 75.8 ± 1.7 | 77.3 ± 1.0 |
| GraphCL pre-trained (You et al., 2020b) | 69.7 ± 0.7 | 75.4 ± 1.4 | 73.9 ± 0.7 | 62.4 ± 0.6 | 60.5 ± 0.9 | 76.0 ± 2.7 | 69.8 ± 2.7 | 78.5 ± 1.2 |
| InfoGraph pre-trained (Wang et al., 2022a) | 66.3 ± 0.6 | 64.8 ± 0.8 | 68.1 ± 0.6 | 58.4 ± 0.6 | 57.1 ± 0.8 | 66.3 ± 0.6 | 44.3 ± 0.6 | 70.2 ± 0.6 |
| JOAOv2 pre-trained (Wang et al., 2022a) | 66.4 ± 0.9 | 67.4 ± 0.7 | 68.2 ± 0.8 | 57.0 ± 0.5 | 59.1 ± 0.7 | 64.5 ± 0.9 | 47.4 ± 0.8 | 68.4 ± 0.5 |
| GraphMAE (Hou et al., 2022) | 72.0 ± 0.6 | 83.1 ± 0.9 | 75.5 ± 0.6 | 64.1 ± 0.3 | 60.3 ± 1.1 | 82.3 ± 1.2 | 76.3 ± 2.4 | 77.2 ± 1.0 |
| GraphLoG (Xu et al., 2021) | 72.5 ± 0.8 | 83.5 ± 1.2 | 75.7 ± 0.5 | 63.5 ± 0.7 | 61.2 ± 1.1 | 76.7 ± 3.3 | 76.0 ± 1.1 | 77.8 ± 0.8 |
| GraphLoG augmented | 65.6 ± 1.0 | 82.5 ± 1.2 | 73.2 ± 0.5 | 63.6 ± 0.4 | 60.9 ± 0.7 | - | - | - |
| LapPE augmented | 67.1 ± 1.6 | 80.4 ± 1.5 | 76.6 ± 0.3 | 65.9 ± 0.7 | 59.3 ± 1.7 | 76.4 ± 2.3 | 75.6 ± 0.8 | 75.6 ± 1.1 |
| RWSE augmented | 67.0 ± 1.4 | 79.6 ± 2.8 | 76.3 ± 0.5 | 65.6 ± 0.3 | 58.5 ± 1.4 | 74.5 ± 4.4 | 75.0 + 1.0 | 78.1 ± 1.5 |
| AllPSE augmented | 67.6 ± 1.2 | 77.0 ± 4.4 | 75.9 ± 1.0 | 63.9 ± 0.3 | **63.0 ± 0.6** | 72.6 ± 4.3 | 67.9 ± 0.7 | 75.4 ± 1.5 |
| GPSE augmented | 66.2 ± 0.9 | 80.8 ± 3.1 | **77.4 ± 0.8** | **66.3 ± 0.8** | 61.1 ± 1.6 | 78.8 ± 3.8 | **76.6 ± 1.2** | 77.2 ± 1.5 |

We note that GPSE augmented GINE achieves the best performance on two out of the five datasets against previously reported performances (Table 4). Moreover, GPSE augmentation improves performance over the baseline across *all* five datasets, unlike some previously reported results that showed negative transfer effects. Together, these results corroborate with the findings from (Sun et al., 2022) that rich features can make up for the benefits of SSL pre-training. In our case, the GPSE encodings act as the rich features that contain positional and structural information from the graphs.

We also highlight that Table 4 results are achieved in a setup where GPSE is at a comparative disadvantage: As a general-purpose feature extractor trained on a separate dataset, GPSE cannot leverage atom and bond features of the downstream graphs unlike the typical molecular graph SSL methods. When GraphLoG is also utilized as a feature extractor for a fairer comparison, we see that it is soundly beaten by GPSE and even suffers from negative transfer despite having access to domain-specific features. This highlights the power of GPSE as a feature extractor. With this in mind, GPSE can instead be combined with other SSL methods to enhance them in future work.

### 3.2 TRANSFERABILITY ACROSS DIVERSE GRAPH BENCHMARKING DATASETS

GPSE can be used on arbitrary types of graphs as it is trained using the graph structures alone, in contrast with the SSL pre-trained methods. Here, we show that GPSE is transferable to general graph datasets apart from molecular datasets, even under extreme out-of-distribution (OOD) cases.

**Transferability to molecular graph sizes** We use the two peptides datasets (Peptides-struct and Peptides-func) from the long-range graph benchmarks (Dwivedi et al., 2022c) to test whether GPSE can still work when the downstream (macro-)molecular graphs are significantly larger than those used for training GPSE. Despite this difference in graph sizes, GPS+GPSE outperforms the original GPS

Table 5: OOD transferability to graph size and connectivity pattern. The GPSE pre-training dataset MolPCBA contains graphs with 26 nodes and 0.093 connectivity on average.

| | Peptides-struct MAE ↓ | Peptides-func AP ↑ | CIFAR10 ACC (%) ↑ | MNIST ACC (%) ↑ |
|---|---|---|---|---|
| Avg. # nodes | 150.9 | 150.9 | 117.6 | 70.6 |
| Avg. connectivity | 0.022 | 0.022 | 0.069 | 0.117 |
| GIN | – | – | 55.26 ± 1.53 | 96.49 ± 0.25 |
| GINE | 0.3547 ± 0.0045 | 0.5498 ± 0.0079 | – | – |
| GatedGCN (Bresson and Laurent, 2017) | 0.3420 ± 0.0013 | 0.5864 ± 0.0077 | 67.31 ± 0.31 | 97.34 ± 0.14 |
| Graph MLP-Mixer (He et al., 2022) | 0.2475 ± 0.0020 | 0.6920 ± 0.0054 | 72.46 ± 0.36 | 98.35 ± 0.10 |
| GPS+(RWSE/LapPE) (Rampášek et al., 2022) | 0.2500 ± 0.0005 | 0.6535 ± 0.0041 | 72.30 ± 0.36 | 98.05 ± 0.13 |
| GPS+AllPSE | 0.2509 ± 0.0028 | 0.6397 ± 0.0092 | 72.05 ± 00.35 | 98.08 ± 00.12 |
| GPS+GPSE | 0.2464 ± 0.0025 | 0.6688 ± 0.0151 | 72.31 ± 0.25 | 98.08 ± 0.13 |

that uses explicitly computed PSEs (Table 5). More strikingly, GPS+GPSE resulted in the new SOTA performance for Peptides-struct, surpassing Graph MLP-Mixer (He et al., 2022). The improved performance due to GPSE emphasizes its ability to better extract global information from query graphs by providing a more informative initial encoding for the global attention mechanism in GPS.

**Transferability to graph connectivity patterns** We further test if GPSE generalizes to graph connectivity patterns distinct from its training dataset. Particularly, we use the superpixel graph benchmarking datasets CIFAR10 and MNIST from Dwivedi et al. (2022a), which are $k$-nearest neighbor graphs with $k = 8$, significantly differing from the molecular graph connectivity patterns. Again, GPS+GPSE achieves comparable results against GPS using explicitly computed PSEs (Table 5).

**Transferability to extreme OOD node-classification benchmarks** Taking the evaluation of GPSE one step further, we test its ability to provide useful information to transductive node classification tasks, where the graphs contain hundreds of thousands of nodes, which are completely out of distribution from the GPSE training dataset. We only use standard baseline methods as a proof of concept, including GCN, GraphSAGE (Hamilton et al., 2017a), and GATv2 (Veličković et al., 2018; Brody et al., 2022). Remarkably, GPSE successfully improves the baseline performance of SAGE and GATv2 on the arXiv dataset, with no noticeable negative transfer (Table 6).

Table 6: OOD transferability to OGB node classification benchmarks.

| | +GPSE? | arXiv ACC (%) ↑ | Proteins AUROC (%) ↑ |
|---|---|---|---|
| GCN | ✗ | 71.74 ± 0.29 | 79.91 ± 0.24 |
| | ✓ | 71.67 ± 0.12 | 79.62 ± 0.12 |
| SAGE | ✗ | 71.74 ± 0.29 | 80.35 ± 0.07 |
| | ✓ | 72.19 ± 0.32 | 80.14 ± 0.22 |
| GAT(E)v2 | ✗ | 71.69 ± 0.21 | 83.47 ± 0.13 |
| | ✓ | 72.17 ± 0.42 | 83.51 ± 0.11 |
| *Best of above* | ✗ | 71.74 ± 0.29 | 83.47 ± 0.13 |
| | ✓ | 72.19 ± 0.32 | 83.51 ± 0.11 |

Meanwhile, the indifference in performance on the Proteins dataset is not unexpected, as the connectivity structures of the protein interaction network do not contribute to the proteins' functions meaningfully. Instead, what matters are the identity of the proteins' interacting partners, commonly referred to as *homophily* in the graph representation learning community (Zhu et al., 2020) or more generally known as the *Guilt-by-Association* principle in the network biology community (Cowen et al., 2017). This result provides valuable insights into the usefulness of GPSE as an augmentation: It is more beneficial when the underlying graph structure is informative for the downstream tasks.

## 3.3 EXPRESSIVENESS OF GPSE ENCODINGS

Given that GPSE can recover different PSEs so well (Table 2.1), it is natural to wonder whether it can boost standard MPNN expressiveness. We first confirm that GPSE encodings surpass 1-WL distinguishability by observing a clear visual separation of GPSE encodings on 1-WL indistinguishable graph pairs (Figure E.1). More information regarding graph isomorphism, the WL test and their connections to GNN expressivity is discussed in Appendix E.

To more rigorously and systematically study the expressivity of GPSE encodings, we perform two synthetic benchmarks (Dwivedi et al., 2022a; Abboud et al., 2021) that require beyond 1-WL power using a 1-WL expressivity bounded MPNN model GIN. Indeed, we find that GPSE provides extra power to the base MPNN model to correctly distinguish graph isomorphism classes (Table 7). This expressivity boosts by GPSE is remarkable, considering that (1) GPSE is pre-trained on MolPCBA, whose graph structures are not designed to be 1-WL indistinguishable like these synthetic graphs, and (2) naively adding random features to the input does not provide such noticeable improvement.

We further point out that, in fact, augmenting the base GIN model using common PSEs like LapPE and RWSE readily archives nearly perfect graph isomorphism classification, corroborating with previous theoretical results on distance-encodings (Li et al., 2020) and spectral invariants (Fürer, 2010). This finding partially explains why GPSE provides additional power and also why previous methods using LapPE achieve perfect classification on these

Table 7: Synthetic graph benchmarks with ten times stratified five-fold CV evaluated in ACC (%) ↑.

| | CSL | | EXP | |
|---|---|---|---|---|
| | Train | Test | Train | Test |
| GIN | 10.0 ± 0.0 | 10.0 ± 0.0 | 49.8 ± 1.8 | 48.7 ± 2.2 |
| GIN+rand | 11.6 ± 3.7 | 12.7 ± 6.4 | 51.0 ± 2.0 | 51.3 ± 2.9 |
| GIN+GPSE | 98.2 ± 1.5 | 42.9 ± 7.9 | 84.6 ± 6.8 | 68.3 ± 7.5 |
| GIN+LapPE | 100.0 ± 0.0 | 92.5 ± 4.2 | 99.9 ± 0.2 | 99.5 ± 0.8 |
| GIN+RWSE | 100.0 ± 0.0 | 100.0 ± 0.0 | 99.7 ± 0.2 | 99.7 ± 0.6 |

tasks (He et al., 2022). Finally, we note the performance difference between LapPE/RWSE and GPSE is not unexpected, as random input features only act as a patch to the MPNN expressivity limitation rather than fully resolving it. Thus, developing powerful and scalable GPSE models that losslessly capture the latent semantics of various PSEs is a valuable venue to explore in the future.

## 3.4 ABLATION STUDIES

**GPSE makes good use of the depth and the global message passing from the VN** Despite the commonly known issue with MPNN oversmoothing as the number of message passing layers increases, we observe that GPSE does not oversmooth thanks to the gating mechanism and the residual connections in GatedGCN (Bresson and Laurent, 2017), contrasting with more typical MPNN layers like GIN (Xu et al., 2019). Indeed, GPSE benefits from both the global message passing by VN and the model depth as shown in Figure 2, confirming our theoretical justifications about the architectural choices in §2.2.

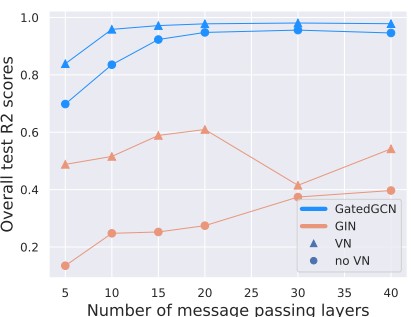

Figure 2: Virtual node, convolution type, and layers ablation using 5% MolPCBA for training GPSE.

**GPSE benefits from the wide variety of pre-training target PSE in downstream tasks** Since GPSE is trained to capture latent semantics for recovering a wide range of PSEs, it mitigates the reliance on manually selecting task-specific PSEs, a major shortcoming of graph Transformers such as GPS and Graphormer (Ying et al., 2021) which could be task specific. For instance, RWSE typically performs well for molecular tasks, while LapPE could be more useful for long-range dependency tasks (Rampášek et al., 2022; Dwivedi et al., 2022c). Here, we investigate whether a particular type of PSE contributes more or less to GPSE by testing the downstream performance of PCQM4Mv2 and MolHIV using different variants of GPSE that excludes one type of PSE during training. We observe from Table F.4 that excluding any type of PSE generally reduces its performance in the downstream tasks slightly, indicating the usefulness of different PSEs' semantics to the downstream tasks at various levels.

**Asymptotic behavior with respect to the GPSE training sample sizes** We perform a scaling law experiment with respect to the training sample sizes, from 5% to 80% of the MolPCBA. As shown in Figure F.1, the testing loss (Appendix B) reduces as the training sample increases. This asymptotic behavior suggests that GPSE can further benefit from the increasing amount of training data.

**The choice of GPSE pre-training dataset affects its downstream performance minimally** We reevaluate the performance of GPSE on PCQM4Mv2 and ZINC when trained on several other choices of molecular graph datasets, including GEOM (Axelrod and Gomez-Bombarelli, 2022), ZINC 250k (Gómez-Bombarelli et al., 2018), PCQM4Mv2 (Hu et al., 2021), and ChEMBL (Gaulton et al., 2012). On ZINC, GPSE performance is variable across different training datasets (Table F.5). Particularly, training GPSE on ChEMBL and MolPCBA, two largest datasets here, results in much better performances than using other, relatively smaller datasets. The superior downstream performance achieved using larger pre-training datasets aligns well with our asymptotic results above, where a larger amount of training samples results in a more accurate GPSE for capturing PSEs, hence leading to better downstream performance. However, we did not observe the same performance difference in the PCQM4Mv2 subset downstream task, indicating that the training size is not always the most crucial factor for good performance, an observation similar to Sun et al. (2022).

Finally, we investigate whether finetuning the GPSE model specifically on the downstream dataset could further improve its downstream performance (Table F.5). Similar to the above findings, we see that further fine-tuning GPSE helps in a task-specific manner, generally providing slight improvements to the ZINC task but less so to the PCQM4Mv2 task. Together, these results of GPSE being minimally affected by different options of pre-training datasets and further fine-tuning to specific downstream datasets reemphasize that GPSE learns *general* and *transferable* knowledge about various PSEs.

## 4 DISCUSSION

**Why does GPSE improve over precomputed PSEs?** Our results demonstrate that GPSE encodings can improve upon augmenting GNNs with precomputed PSEs in downstream tasks. The fact that we can *recover* the target PSEs in pretraining (Table 1) accounts for why we can match the original PSEs. Why we *improve* upon them, meanwhile, can be attributed to our joint encoding: Learning to encode a diverse collection of PSEs leads to a general embedding space that abstracts both local and global perspectives of the query graph, which are more readily useable by the downstream model compared to the unprocessed PSEs.

**What advantages does GPSE bring over traditional SSL pre-training?** In addition to being less prone to negative transfer and having competitive performance (Table 4), GPSE provides a few more advantages over traditional SSL pre-training methods: (1) GPSE uses randomly generated features instead of dataset-specific graph features, thus can be applied to arbitrary graph datasets; (2) GPSE is only used as a feature extractor and hence does not impose any constraint on the downstream model. Despite these differences, we emphasize that GPSE can be complementary to traditional SSL to further enhance the prediction performance, for example, by using GPSE encodings as input features to the SSL pre-training.

**Why is GPSE transferable to OOD data?** The transferability of GPSE to OOD data is uncommon in the graph SSL pre-training literature, particularly for applications with molecular graphs. We hypothesize that GPSE's transferability is a consequence of the choice of its predictive self-supervision tasks, that contain a mixture of both local and global intrinsic graph information. This encourages GPSE to capture global invariances using local information, hence allowing it to extract valuable representations on graphs that are different in sizes and connectivity from the training graphs.

**When does GPSE help, and when does it not?** GPSE provides essential information to the model when the downstream task requires positional or structural information of the graph or better node identifiability in general, which is typically the case for molecular property predictions (Hu et al., 2020b). Conversely, for downstream tasks that do not rely on such information, e.g. protein function prediction using the protein interaction network (Table 6), the benefits from GPSE are not as apparent.

## 5 CONCLUSION

We have introduced the GPSE, a unifying graph positional and structural encoder for augmenting any graph learning dataset while being applicable to all graph Transformers and message-passing GNN models. GPSE extracts rich node encodings by learning to predict a diverse collection of predefined PSEs in the initial self-supervised training phase on a set of unattributed graphs. We demonstrated a superior performance of GPSE encodings over the explicitly constructed PSEs on a variety of graph learning benchmarks. Furthermore, GPSE showed great transferability across diverse benchmarking datasets and even achieved a new SOTA performance on the Peptides-struct long-range benchmark, whose graph structures are significantly different from those in the MolPCBA dataset, that was used to train the GPSE. Our study opens up exciting opportunities for learning a graph encoder as a unified PSE to augment GNN models, and we hope our work will motivate future studies toward learning even more powerful graph PSE encoders to advance graph analysis.

**Limitations and future directions** Despite the effectiveness of our GPSE model design choices, it is currently prohibitively large to be trained on graph datasets with over one million graphs efficiently. As GPSE asymptotically achieves perfect PSE recovery, it is a promising future direction to make GPSE more efficient and thus allow it to be trained on billion scale molecular graph datasets (Patel et al., 2020; Irwin et al., 2020).

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

## A  POSITIONAL AND STRUCTURAL ENCODING TASKS DETAIL

We consider a simple undirected and unweighted graph $G = (V, E)$ as a tuple of the vertex set $V$ and the edge set $E$, with no node or edge features. We denote the number of nodes and the number of edges as $n = |V|$ and $m = |E|$, respectively. Then, the corresponding adjacency matrix representing the graph $G$ is a symmetric matrix $\mathbf{M} \in \{0,1\}^{n \times n}$, where $\mathbf{M}_{ij} = 1$ if $(v_i, v_j) \in E$ and 0 otherwise. The graph Laplacian $\mathbf{L}$ is defined as

$$\mathbf{L} = \mathbf{D} - \mathbf{M} \tag{A.1}$$

where $\mathbf{D} \in \mathbb{N}^{n \times n}$ is a diagonal matrix whose entries correspond to the degree of a vertex in the graph, $\mathbf{D}_{ii} = deg(v_i) = |\mathcal{N}(v_i)| = |\{u|(v_i, u) \in E\}|$.

The graph Laplacian is a real symmetric matrix, thus having a full eigendecomposition as

$$\mathbf{L} = \mathbf{U}\mathbf{\Lambda}\mathbf{U}^\top \tag{A.2}$$

where, $\mathbf{\Lambda}_{ii} = \lambda_i$ and $\mathbf{U}_{[:,i]} = u_i$ are the $i^{\text{th}}$ eigenvalue and eigenvector (an eigenpair) of the graph Laplacian. We follow the convention of indexing the eigenpair from the smallest to the largest eigenvalue, i.e., $0 = \lambda_1 \leq \lambda_2 \leq \cdots \leq \lambda_n$. We further denote $\hat{\mathbf{U}}$ (and analogously the subdiagonal matrix $\hat{\mathbf{\Lambda}}$) as the matrix of Laplacian eigenvectors corresponding to non-trivial eigenvalues.

$$\hat{\mathbf{U}} = \mathbf{U}_{[:,\{i|\lambda_i \neq 0\}]} \tag{A.3}$$

Finally, we denote the ($\ell_2$) normalization operation as $normalize(x) := \frac{x}{\|x\|_2}$

**Laplacian eigenvector positional encodings (LapPE)**   LapPE is defined as the absolute value of the $\ell_2$ normalized eigenvectors associated with non-trivial eigenvalues. We use the first four LapPE to train GPSE by default.

$$\text{LapPE}_i = |normalize(\hat{\mathbf{U}}_{[:,i]})| \tag{A.4}$$

The absolute value operation is needed to counter the sign ambiguity of the graph Laplacian eigenvectors, a known issue to many previous works that use the Laplacian eigenvectors to augment the models Dwivedi et al. (2022a); Lim et al. (2022). However, common strategies to overcome the sign ambiguity issue such as random sign flipping Dwivedi et al. (2022a) or constructing sign invariant function Lim et al. (2022) do not resolve our issue here as we are trying to *recover* the PEs rather than using them as features. Investigating the strategies for learning *invariant* representations for eigenvectors could be an interesting venue for future studies.

We additionally use the eigenvalues as a graph-level regression task for training GPSE.

**Electrostatic potential positional encodings (ElstaticPE)**   The pseudoinverse of the graph Laplacian $\mathbf{L}^\dagger$ has a physical interpretation that closely relates to the electrostatic potential between two nodes in the graph $G$ when each node is treated as a charged particle  Kreuzer et al. (2021) and can be computed as

$$\mathbf{L}^\dagger = \mathbf{U}\mathbf{\Lambda}^\dagger\mathbf{U}^\top = \hat{\mathbf{U}}\hat{\mathbf{\Lambda}}^{-1}\hat{\mathbf{U}}^\top \tag{A.5}$$

We further subtract each column of $\mathbf{L}^\dagger$ by its diagonal value to set zero ground state such that each node's potential on itself is 0.

$$\mathbf{Q} = \mathbf{L}^\dagger - diag(\mathbf{L}^\dagger)\mathbf{1}_n \tag{A.6}$$

The final ElstaticPE is a collection of aggregated values for each node, that summarizes the electrostatic interaction of a node with all other nodes :

1. Minimum potential from $v_i$ to $v_j$: $\text{ElstaticPE}_1(i) = \min(\mathbf{Q}_{[:,i]})$

2. Average potential from $v_i$ to $v_j$: $\text{ElstaticPE}_2(i) = \text{mean}(\mathbf{Q}_{[:,i]})$

3. Standard deviation of potential from $v_i$ to $v_j$: $\text{ElstaticPE}_3(i) = \text{std}(\mathbf{Q}_{[:,i]})$

4. Minimum potential from $v_j$ to $v_i$: $\text{ElstaticPE}_4(i) = \min(\mathbf{Q}_{[i,:]})$

5. Standard deviation of potential from $v_j$ to $v_i$: $\text{ElstaticPE}_5(i) = \text{std}(\mathbf{Q}_{[i,:]})$

6. Average interaction on direct neighbors: $\text{ElstaticPE}_6(i) = \text{mean}\left((\mathbf{MQ})_{[:,i]}\right)$

7. Average interaction from direct neighbors: $\text{ElstaticPE}_7(i) = \text{mean}\left((\mathbf{MQ})_{[i,:]}\right)$

**Random walk structural encodings (RWSE)** Define the random walk matrix as the row-normalized adjacency matrix $\mathbf{P} := \mathbf{D}^{-1}\mathbf{M}$. Then $\mathbf{P}_{i,j}$ corresponds to the one-step transition probability from $v_i$ to $v_j$.

The $k^{\text{th}}$ RWSE Dwivedi et al. (2022b) is defined as the probability of returning back to the starting state of a random walk after exactly $k$ step of random walks:

$$\text{RWSE}_k = diag(\mathbf{P}^k) \tag{A.7}$$

**Heat kernel diagonal structural encodings (HKdiagSE)**

$$\text{HKdiagSE}_k = \sum_{i:\lambda_i \neq 0} e^{-k\lambda_i} normalize(\mathbf{U}_{[:,i]})^2 \tag{A.8}$$

**Cycle counting structural encodings (CycleSE)** CycleSE encodes global structural information of the graph by counting the number of $k$-cycles in the graph. For example, a 2-cycle corresponds to an undirected edge, and a 3-cycle corresponds to a triangle.

$$\text{CycleSE}_k = |\{\text{Cycles of length k}\}| \tag{A.9}$$

CycleSE is used as a graph-level regression task for training GPSE.

**Normalizing PSEs tasks** Finally, we perform graph-wide normalization preprocessing step on each node-level PSE task so that they have zero mean and unit standard deviation. This normalization step ensures all PSE targets are on the same scale, making the training process more stable.

# B    IMPLEMENTATION DETAILS

## B.1    GPSE COMPUTATION

The GPSE model is built using a GatedGCN backbone (Bresson and Laurent, 2018) with PSE-specific MLP decoding heads. GPSE uses random noise drawn from a 20-dimensional standard Gaussian as the input node features. The random features are then projected to the match the hidden dimension, $d$, of the model, resulting in the hidden representations of the first layer:

$$h_i^{(0)} = \text{ReLU}\left( x_i W_{\text{inp}} \right) \tag{B.1}$$

where $h_i^{(0)} \in \mathbb{R}^{1 \times d}$ indicates the hidden feature of node $i$ in the first layer, $W_{\text{inp}} \in \mathbb{R}^{20 \times d}$ is the linear projection layer, and $x_i \sim \mathcal{N}(\mathbf{0}, \mathbf{I}) \in \mathbb{R}^{1 \times 20}$ is the random noise. Next, the model enters $L$ layers of GatedGCN convolution layers, where each layer is define as:

$$h_i^{(l+1)} = \text{ReLU}\left( h_i^{(l)} W_1^{(l)} + \sum_{j \in \mathcal{N}(i)} \sigma\left( h_i^{(l)} W_2^{(l)} + h_j^{(l)} W_3^{(l)} \right) \odot \left( h_j^{(l)} W_4^{(l)} \right) \right) \tag{B.2}$$

where $W_1^{(l)}, W_2^{(l)}, W_3^{(l)}, W_4^{(l)} \in \mathbb{R}^{d \times d}$ are learnable parameters for layer $l$, $\sigma$ is the sigmoid function, and $\odot$ is the elementwise multiplication operator. Finally, the processed hidden feature $h_i^{(L)}$ is decoded via a two-layer MLP to predict the $k^{\text{th}}$ the node-level PSEs, such as LapPE and RWSE.

$$\hat{y}_{i,k} = \text{ReLU}\left( h_i^{(L)} W_{k,1} \right) W_{k,2} \tag{B.3}$$

where $W_{k,1} \in \mathbb{R}^{d \times d}$ and $W_{k,2} \in \mathbb{R}^{d \times 1}$ are learnable parameters for projecting the final hidden representation to the PSE prediction. For graph-level PSEs, such as CycleSE, we use sum-pooling to reduce the hidden representations to graph-level first, and similarly apply a two-layer MLP afterwards. Once trained, we apply GPSE to extract $h^{(L)}$ for the graphs in the downstream dataset and use it in-place of the traditional PSEs. We set $L$ to 20, and $d$ to 512 for our final GPSE architecture. We also present an ablation study on various architectural choices to demonstrate the effectiveness of our final model setting (Table F.1).

## B.2    GPSE TRAINING LOSS FUNCTION

We use a combination of $\ell_1$ loss and cosine similarity loss for training GPSE using the PSE self-supervision defined in Appendix A. More specifically, given $M$ number of graphs, and $K$ number of target PSE tasks, we compute the loss as follows:

$$\mathcal{L} = \sum_{k=1}^{K} \sum_{i=1}^{M} \left[ \left( \sum_{j=1}^{|V(G_i)|} \left| y_{j,k}^{(i)} - \hat{y}_{j,k}^{(i)} \right| \right) + \left( 1 - \sum_{j=1}^{|V(G_i)|} \tilde{y}_{j,k}^{(i)} \tilde{\hat{y}}_{j,k}^{(i)} \right) \right] \tag{B.4}$$

where $y_{j,k}^{(i)}, \hat{y}_{j,k}^{(i)}$ are the true and predicted values of the $j^{\text{th}}$ node of $i^{\text{th}}$ graph for the $k^{\text{th}}$ PSE task. $\tilde{y}$ and $\tilde{\hat{y}}$ are the $\ell_2$ normalized version of $y$ and $\hat{y}$, respectively. Note that in practice, we compute the loss over mini-batches of graphs rather than over all of the training graphs.

## B.3    COMPUTE ENVIRONMENT AND RESOURCES

Our codebase is based on GraphGPS Rampášek et al. (2022), which uses PyG and its GraphGym module Fey and Lenssen (2019); You et al. (2020a). All experiments are run using Tesla V100 GPUs (32GB), with varying numbers of CPUs from 4 to 8 and up to 48GB of memory (except for two cases: (i) 80GB of memory is needed when performing downstream evaluation on MolPCBA, and (ii) 128GB is needed when pre-training GPSE on the ChEMBL dataset).

We recorded the run time for both the GPSE pre-computation and the downstream evaluation training loop using Python's `time.perf_counter()` function and reported them in Table B.1, B.2, B.3. We did not report the GPSE pre-computation time for other downstream benchmarks since they are all within five minutes.

## B.4 HYPERPARAMETERS

### B.4.1 DOWNSTREAM TASKS

In most of the downstream task hyperparameter searches, we followed the best settings from previous studies Rampášek et al. (2022), and primarily tuned the GPSE encoding parameters, including the GPSE processing encoder type, the encoded dimensions, the input and output dropout rate of the processing encoder, and the application of batch normalization to the input GPSE encodings. For completeness, we list all hyperparameters for our main benchmarking studies in Table B.1, B.2.

Table B.1: GPS+GPSE hyperparameters for molecular property prediction benchmarks

| Hyperparameter | ZINC (subset) | PCQM4Mv2 (subset) | MolHIV | MolPCBA |
|---|---|---|---|---|
| # GPS Layers | 10 | 5 | 10 | 5 |
| Hidden dim | 64 | 304 | 64 | 384 |
| GPS-MPNN | GINE | GatedGCN | GatedGCN | GatedGCN |
| GPS-SelfAttn | – | Transformer | Transformer | Transformer |
| # Heads | 4 | 4 | 4 | 4 |
| Dropout | 0.00 | 0.00 | 0.05 | 0.20 |
| Attention dropout | 0.50 | 0.50 | 0.50 | 0.50 |
| Graph pooling | mean | mean | mean | mean |
| PE dim | 32 | 128 | 20 | 48 |
| PE encoder | 2-Layer MLP | 2-Layer MLP | Linear | Linear |
| Input dropout | 0.50 | 0.50 | 0.30 | 0.30 |
| Output dropout | 0.00 | 0.20 | 0.10 | 0.10 |
| Batchnorm | yes | no | yes | yes |
| Batch size | 32 | 256 | 32 | 512 |
| Learning rate | 0.001 | 0.0002 | 0.0001 | 0.0005 |
| # Epochs | 2000 | 100 | 100 | 100 |
| # Warmup epochs | 50 | 5 | 5 | 5 |
| Weight decay | 1.00e-5 | 1.00e-6 | 1.00e-5 | 1.00e-5 |
| # Parameters | 292,513 | 6,297,345 | 573,025 | 9,765,264 |
| PE precompute | 2 min | 1.5 hr | 8 min | 1.3 hr |
| Time (epoch/total) | 10s/5.78h | 102s/2.82h | 121s/3.37h | 185s/5.15h |

**MoleculeNet small benchmarks settings** We used the default GINE architecture following previous studies Hu et al. (2020b), which has five hidden layers and 300 hidden dimensions. For all five benchmarks, we use the same GPSE processing encoder settings as shown in Table B.4a.

**CSL & EXP synthetic graph benchmarks settings** We follow He et al. (2022) and use GIN Xu et al. (2019) as the underlying MPNN model, with five hidden layers and 128 dimensions. We use the same GPSE processing encoder settings for both CSL and EXP as shown in Table B.4b.

Table B.2: GPS+GPSE hyperparameters for transferability benchmarks

| Hyperparameter | Peptides-struct | Peptides-func | CIFAR10 | MNIST |
|---|---|---|---|---|
| # GPS Layers | 4 | 4 | 3 | 3 |
| Hidden dim | 96 | 96 | 52 | 52 |
| GPS-MPNN | GatedGCN | GatedGCN | GatedGCN | GatedGCN |
| GPS-SelfAttn | Transformer | Transformer | Transformer | Transformer |
| # Heads | 4 | 4 | 4 | 4 |
| Dropout | 0.00 | 0.00 | 0.00 | 0.00 |
| Attention dropout | 0.50 | 0.50 | 0.50 | 0.50 |
| Graph pooling | mean | mean | mean | mean |
| PE dim | 8 | 24 | 8 | 8 |
| PE encoder | Linear | 2-Layer MLP | 2-Layer MLP | Linear |
| Input dropout | 0.10 | 0.10 | 0.30 | 0.50 |
| Output dropout | 0.05 | 0.00 | 0.00 | 0.00 |
| Batchnorm | yes | yes | no | no |
| Batch size | 128 | 128 | 16 | 16 |
| Learning rate | 0.0005 | 0.0003 | 0.001 | 0.001 |
| # Epochs | 200 | 200 | 100 | 100 |
| # Warmup epochs | 10 | 10 | 5 | 5 |
| Weight decay | 1.00e-4 | 0 | 1.00e-5 | 1.00e-4 |
| # Parameters | 510,435 | 529,250 | 120,886 | 119,314 |
| PE precompute | 3 min | 3 min | 14 min | 16 min |
| Time (epoch/total) | 12s/0.65h | 12s/0.67h | 88s/2.44h | 104s/2.90h |

Table B.3: Downstream MPNN hyperparameters for node-level benchmarks.

| Hyperparameter | arXiv | Proteins |
|---|---|---|
| MPNN | SAGE | GATEv2 |
| # MPNN Layers | 3 | 3 |
| Hidden dim | 256 | 256 |
| Dropout | 0.50 | 0.00 |
| PE dim | 32 | 32 |
| PE encoder | 2-Layer MLP | Linear |
| Input dropout | 0.50 | 0.40 |
| Output dropout | 0.20 | 0.05 |
| Batchnorm | no | no |
| Learning rate | 0.01 | 0.01 |
| # Epochs | 500 | 1000 |
| Weight decay | 0 | 0 |
| # Parameters | 534,888 | 910,448 |
| PE precompute | 8 sec | 3 min |
| Time (epoch/total) | 0.25s/0.07h | 35s/9.40h |

Table B.4: `GPSE` processing encoder hyperparameters for MoleculeNet small benchmarks and synthetic WL graph benchmarks

(a) MoleculeNet small benchmarks settings

| Hyperparameter | |
| --- | --- |
| PE dim | 64 |
| PE encoder | Linear |
| Input dropout | 0.30 |
| Output dropout | 0.10 |
| Batchnorm | yes |
| Learning rate | 0.003 |
| # Epochs | 100 |
| # Warmup epochs | 5 |
| Weight decay | 0 |

(b) Synthetic WL graph benchmarks settings

| Hyperparameter | |
| --- | --- |
| PE dim | 128 |
| PE encoder | Linear |
| Input dropout | 0.00 |
| Output dropout | 0.00 |
| Batchnorm | yes |
| Learning rate | 0.002 |
| # Epochs | 200 |
| Weight decay | 0 |

## C   THEORY DETAILS

Message-passing GNNs have receptive fields that grow exponentially with the number of layers. Given two nodes, the influence of one onto the other might become too weak over long graph distances, hindering the learning task. This phenomenon has been referred to as *over squashing* Alon and Yahav (2021). A similar problem also occurs as the number of layers increases, where the nodes' hidden representations become increasingly similar as the number of layers increase: A phenomenon commonly referred to as *over-smoothing* Li et al. (2019).

### C.1   RELEVANCE TO GPSE

The over-smoothing and over-squashing problems are essential to overcome to effectively learn the positional and structural encodings, especially for those that require *global views of the graph*. For example, the Laplacian eigenvector corresponding to the first non-trivial eigenvalue, also known as the Fiedler vector, corresponds to the solution of the graph min-max cut problem Ding et al. (2001). Intuitively, this problem requires accessing the global view of the entire graph as it, colloquially, aims to partition the entire graph into two parts with minimal connections.

A straightforward solution to incorporating more global information into the model is by *stacking more message-passing layers* to increase the receptive field and thus effectively expose the model to information beyond the local structure. However, simply stacking more message-passing layers easily leads to the *over-smoothing* problem, where the messages of each node become increasingly uniform as the number of layers increases. Our usage of the gating mechanism, along with residual connection, effectively mitigates this issue while still exposing the model to more non-local information.

Meanwhile, the model may still have difficulty incorporating global information, even after fixing the over-smoothing issue and stacking more layers due to *over-squashing*. Informally, over-squashing can be understood as the difficulty in losslessly sending messages between two nodes across the network. This difficulty is primarily because there are only a few possible routes between the two nodes compared to all other available routes to each of the nodes. We mitigate this problem using a *virtual node* that serves as the global information exchange hub to enable global information exchange, bypassing the "few routes" limitation.

### C.2   FORMAL ANALYSIS

**Definition 1** (Over-squashing). *The squashing of a GNN is measured by the influence of one node on the features of another which we interpret as the partial derivative*

$$\frac{\partial h_i^{(r+1)}}{\partial x_j}$$

*for $h_i^{(r)}(x_1, ..., x_n)$ the $r$-th hidden feature at node $i$, and $x_j$ the input feature at node $j$. If this quantity converges to 0 as $r$ increases, then the network is said to suffer from* over-squashing.

Another common problem with MPNNs is known as *over-smoothing*. It has often been observed that MPNNs with many layers produce node features that are very close or even identical, which limits expressivity and prevents learning. This stems from message-passing being equivalent to a local smoothing operation; too many smoothing iterations result in all nodes converging to identical states.

**Definition 2** (Over-smoothing). *The smoothing of a network can be measured by the norm (for example the $\ell_1$-norm) of the state difference between neighbors, i.e.*

$$\sum_{(i,j) \in E} |h_i^{(r)} - h_j^{(r)}|$$

*where the sum is taken over the edges of the graph. If this quantity converges to 0 as $r$ increases, the network is said to suffer from* over-smoothing.

In the following section, we will refer to the relationships between over-squashing and over-smoothing with graph curvature. There are many versions of graph curvature Forman (2003); Ollivier (2009); Sreejith et al. (2016); Topping et al. (2022), all closely related. Here we will only consider the balanced Forman curvature from Topping et al. (2022).

**Definition 3** (Graph curvature). *For any edge $(i, j)$ in a simple, unweighted graph $G$, its contribution to graph curvature is given by*

$$\mathrm{Ric}(i, j) = \frac{2}{d_i} + \frac{2}{d_j} - 2 + |\#_\triangle(i, j)| \left( \frac{2}{\max\{d_i, d_j\}} + \frac{1}{\min\{d_i, d_j\}} \right) + \frac{\gamma_{\max}}{\max\{d_i, d_j\}} \left( |\#_\square^i| + |\#_\square^j| \right)$$

*where $\#_\square^i$ is the number of 4-cycles containing the node $i$ (diagonals not allowed), $\#_\triangle^i$ is the number of 3-cycles containing $i$, $d_i$ is the degree of $i$ and $\gamma_{max}$ is the maximum over nodes $k$ of the number of 4-cycles that pass through the nodes $i, j$ and $k$.*

It can then be shown that negative curvature causes over squashing Topping et al. (2022); Nguyen et al. (2022) and positive curvature causes over smoothing Nguyen et al. (2022); Giraldo et al. (2022).

Next, we show that rewiring the graph by adding a virtual node increases the balanced Forman curvature of the graph at most edges.

**Proposition 1.** *The balanced Forman Curvature is increased for most edges when adding a virtual node such that*

$$\mathrm{Ric}(i, j) - \mathrm{Ric}^{+\mathrm{VN}}(i, j) \leq \frac{1}{(d_i - \delta)^2 + d_i - \delta} - \frac{2\delta}{d_i^2 + d_i},$$

*where $d_i$ is the degree of the most connected node of the edge $(i, j)$ and $\delta = d_i - d_j$.*

**Proof.** $\#_\square$ is invariant when adding virtual node because it automatically creates diagonals in the new 4-cycles. Therefore, $\gamma_{\max}$ is also invariant. As for $d_i$, $d_j$ and $\#_\triangle$, they are all increased by 1:

$$\mathrm{Ric}(i, j) - \mathrm{Ric}^+(i, j) \approx 2 \left( \frac{1}{d_i} + \frac{1}{d_j} - \frac{1}{d_i + 1} - \frac{1}{d_j + 1} \right) + |\#_\triangle(i, j)| \left( \frac{2}{\max\{d_i, d_j\}} + \frac{1}{\min\{d_i, d_j\}} \right)$$

$$- \left( |\#_\triangle(i, j)| + 1 \right) \left( \frac{2}{\max\{d_i, d_j\} + 1} + \frac{1}{\min\{d_i, d_j\} + 1} \right)$$

We can let $d_i \geq d_j$ without loss of generality. The inequality is not influenced by the introduction of a virtual node:

$$\mathrm{Ric}(i, j) - \mathrm{Ric}^+(i, j) \approx 2 \left( \frac{1}{d_i^2 + d_i} + \frac{1}{d_j^2 + d_j} \right) + |\#_\triangle(i, j)| \left( \frac{2}{d_i} + \frac{1}{d_j} \right) - (|\#_\triangle(i, j)| + 1) \left( \frac{2}{d_i + 1} + \frac{1}{d_j + 1} \right)$$

$$\mathrm{Ric}(i, j) - \mathrm{Ric}^+(i, j) \approx 2 \left( \frac{1}{d_i^2 + d_i} + \frac{1}{d_j^2 + d_j} \right) + |\#_\triangle(i, j)| \left( \frac{2}{d_i^2 + d_i} + \frac{1}{d_j^2 + d_j} \right) - \left( \frac{2}{d_i + 1} + \frac{1}{d_j + 1} \right)$$

The number of triangles is upper bounded by the least connected node's degree minus 1, $|\#_\triangle| \leq d_j - 1$. We then have:

$$\mathrm{Ric}(i, j) - \mathrm{Ric}^+(i, j) \leq 2 \left( \frac{1}{d_i^2 + d_i} + \frac{1}{d_j^2 + d_j} \right) + (d_j - 1) \left( \frac{2}{d_i^2 + d_i} + \frac{1}{d_j^2 + d_j} \right) - \left( \frac{2}{d_i + 1} + \frac{1}{d_j + 1} \right)$$

$$= -\frac{2(d_i - 1)}{d_i^2 + d_i} - \frac{d_j - 2}{d_j^2 + d_j} + (d_j - 1) \left( \frac{2}{d_i^2 + d_i} + \frac{1}{d_j^2 + d_j} \right)$$

$$= -\frac{2(d_i - 1 - d_j + 1)}{d_i^2 + d_i} - \frac{d_j - 2 - d_j + 1}{d_j^2 + d_j}$$

$$= \frac{1}{d_j^2 + d_j} - \frac{2(d_i - d_j)}{d_i^2 + d_i}$$

Let's call the difference between the two nodes' degrees $\delta$. We get:

$$\mathrm{Ric}(i, j) - \mathrm{Ric}^+(i, j) \leq \frac{1}{(d_i - \delta)^2 + d_i - \delta} - \frac{2\delta}{d_i^2 + d_i}$$

This upper bound gives us a good insight on the general behavior of the curvature when adding a virtual node.

**First case:** The upper bound of the difference is negative for $\delta \neq 0$ and $d_j \neq 1$. This means that for the most cases, the addition of the virtual node clearly increases the curvature.

**Second case:** The upper bound is positive for $d_j = 1$ (ie. $d_i - \delta = 1$). However, the isolated edges are not responsible for bottleneckness. It is to be noted that such *isolated* edges never have negative curvature, neither before nor after the addition of the virtual node. This is a direct consequence of the curvature definition.

**Third case:** The upper bound is positive for $\delta = 0$. This comes as a surprise and might need futur work. It is to be noted that the upper bound tends toward 0 pretty quickly as (as $\frac{1}{d_i^2}$), thus, by the nature of the upper bound, the addition of the virtual node should still increase curvature for most of the cases where $\delta = 0$.

**Note** that we didn't include the 4-cycle term, because this term is inversely proportional to the number of triangles, and is therefore equal to 0 when the number of triangle is maximal. Otherwise, as the number of triangle decreases, the upper bound on the 4-cycle term increases, in a slower fashion. Thus, the upper bound still holds.

# D DATASETS

Table D.1: Task information for datasets used in transferability experiments.

| Dataset | Num. graphs | Num. nodes | Num. edges | Pred. level | Pred. task | Num. tasks | Metric |
|---|---|---|---|---|---|---|---|
| ZINC-subset | 12,000 | 23.15 | 24.92 | graph | reg. | 1 | MAE |
| CIFAR10 | 60,000 | 117.63 | 469.10 | graph | class. (10-way) | 1 | ACC |
| MNIST | 70,000 | 70.57 | 281.65 | graph | class. (10-way) | 1 | ACC |
| MolHIV | 41,127 | 25.51 | 27.46 | graph | class. (binary) | 1 | AUROC |
| MolPCBA | 437,929 | 25.97 | 28.11 | graph | class. (binary) | 128 | AP |
| MolBBBP | 2,039 | 24.06 | 25.95 | graph | class. (binary) | 1 | AUROC |
| MolBACE | 1,513 | 34.09 | 36.86 | graph | class. (binary) | 1 | AUROC |
| MolTox21 | 7,831 | 18.57 | 19.29 | graph | class. (binary) | 21 | AUROC |
| MolToxCast | 8,576 | 18.78 | 19.26 | graph | class. (binary) | 617 | AUROC |
| MolSIDER | 2,039 | 33.64 | 35.36 | graph | class. (binary) | 27 | AUROC |
| PCQM4Mv2-subset | 446,405 | 14.15 | 14.58 | graph | reg. | 1 | MAE |
| Peptides-func | 15,535 | 150.94 | 153.65 | graph | class. (binary) | 10 | AP |
| Peptides-struct | 15,535 | 150.94 | 153.65 | graph | reg. | 11 | MAE |
| CSL | 150 | 41.00 | 82.00 | graph | class. (10-way) | 1 | ACC |
| EXP | 1,200 | 48.70 | 60.44 | graph | class. (binary) | 1 | ACC |
| arXiv | 1 | 169K | 40M | node | class. (40-way) | 1 | ACC |
| Proteins | 1 | 133K | 1.2M | node | class. (binary) | 112 | AUROC |

Table D.2: Classical graph properties of graph-level datasets used in transferability experiments.

| | Num. nodes | Num. edges | Density | Connectivity | Diameter | Approx. max clique | Centrality | Cluster. coeff. | Num. triangles |
|---|---|---|---|---|---|---|---|---|---|
| ZINC-subset | 23.15 | 24.92 | 0.101 | 1.00 | 12.47 | 2.06 | 0.101 | 0.006 | 0.06 |
| CIFAR10 | 117.63 | 469.10 | 0.068 | 3.56 | 9.14 | 5.65 | 0.068 | 0.454 | 502.66 |
| MNIST | 70.57 | 281.65 | 0.116 | 3.71 | 6.83 | 5.56 | 0.116 | 0.478 | 316.65 |
| MolHIV | 25.51 | 27.46 | 0.103 | 0.927 | 11.06 | 2.02 | 0.103 | 0.002 | 0.03 |
| MolPCBA | 25.97 | 28.11 | 0.093 | 0.998 | 13.56 | 2.02 | 0.093 | 0.002 | 0.02 |
| MolBBBP | 24.06 | 25.95 | 0.114 | 0.950 | 10.75 | 2.03 | 0.114 | 0.003 | 0.03 |
| MolBACE | 34.09 | 36.86 | 0.070 | 1.00 | 15.22 | 2.10 | 0.070 | 0.007 | 0.10 |
| MolTox21 | 18.57 | 19.29 | 0.157 | 0.976 | 9.37 | 2.02 | 0.159 | 0.003 | 0.03 |
| MolToxCast | 18.78 | 19.26 | 0.154 | 0.803 | 7.57 | 2.02 | 0.154 | 0.003 | 0.03 |
| MolSIDER | 33.64 | 35.36 | 0.103 | 0.856 | 12.45 | 2.02 | 0.120 | 0.004 | 0.04 |
| PCQM4Mv2-subset | 14.15 | 14.58 | 0.163 | 1.00 | 7.95 | 2.06 | 0.163 | 0.010 | 0.07 |
| Peptides-func | 150.94 | 153.65 | 0.022 | 0.990 | 56.42 | 2.00 | 0.022 | 0.000 | 0.001 |
| Peptides-struct | 150.94 | 153.65 | 0.022 | 0.990 | 56.42 | 2.00 | 0.022 | 0.000 | 0.001 |
| CSL | 41.00 | 82.00 | 0.100 | 3.98 | 6.00 | 2.10 | 0.100 | 0.050 | 4.10 |
| EXP | 48.70 | 60.44 | 0.054 | 0.00 | 1.00 | 2.00 | 0.054 | 0.000 | 0.00 |

## D.1 PRE-TRAINING DATASETS

**MolPCBA** Hu et al. (2020a) (MIT License) contains 400K small molecules derived from the MoleculeNet benchmark Wu et al. (2018). There are 323,555 unique molecular graphs in this dataset.

**ZINC** Gómez-Bombarelli et al. (2018) (Apache 2.0 License) contains 250K drug-like commercially available small molecules sampled from the full ZINC Irwin et al. (2012) database. There are 219,2384 unique molecular graphs in this dataset.

**GEOM** Axelrod and Gomez-Bombarelli (2022) (CC0 1.0 license) consists of 300K drug-like small molecules. There are 169,925 unique molecular graphs in this dataset.

**ChEMBL** Gaulton et al. (2012)[1] (CC BY-SA 3.0 License) consists of 1.4M drug-like bioactive small molecules. There are 970,963 unique molecular graphs in this dataset.

**PCQM4Mv2** Hu et al. (2021) (CC BY 4.0 License) contains 3.4M small molecules from the PubChemQC Nakata and Shimazaki (2017) project. The ground-state electronic structures of these

---

[1]We used release 32 of ChEMBL: http://doi.org/10.6019/CHEMBL.database.32

molecules were calculated using Density Functional Theory. There are 273,920 unique molecular graphs in this dataset.

### D.1.1 EXTRACTING UNIQUE MOLECULAR GRAPH STRUCTURES

To extract unique molecular graphs, we use RDKit with the following steps:

1. For each molecule, convert all its heavy atoms to carbon and all its bonds to single-bond.
2. Convert the modified molecules into a list of SMILES strings.
3. Reduce the list to unique SMILES strings using the `set()` operation in Python.

### D.2 DOWNSTREAM EVALUATION DATASETS

**ZINC-subset** Dwivedi et al. (2022a) is a 12K subset of the ZINC250K dataset Gómez-Bombarelli et al. (2018). Each graph is a molecule whose nodes are atoms (28 possible types) and whose edges are chemical bonds (3 possible types). The goal is to regress the constrained solubility Dwivedi et al. (2022a) (logP) of the molecules. This dataset comes with a pre-defined split with 10K training, 1K validation, and 1K testing samples.

**MolHIV & MolPCBA** Hu et al. (2020a) (MIT License) are molecular property prediction datasets derived from the MoleculeNet benchmarks Wu et al. (2018). Each graph represents a molecule whose nodes are atoms (9-dimensional features containing atom type, chirality, etc.) and whose edges are chemical bonds. The goal for MolHIV is to predict molecules' ability to inhibit HIV virus replication as a binary classification task. On the other hand, MolPCBA consists of 128 binary classification tasks that are derived from high-throughput bioassay measurements. Both datasets come with pre-defined splits based on the *scaffold splitting* procedure Hu et al. (2020b).

**PCQM4Mv2-subset** Hu et al. (2021); Rampášek et al. (2022) (CC BY 4.0 License) is a subsampled version of PCQM4Mv2 Hu et al. (2021) using random 10% for training, 33% for validation, and the original *validatation* set for testing. The molecular graphs are processed the same way as for MolHIV and MolPCBA, where each node is an atom, and each edge is a chemical bond. The task is to regress the HOMO-LUMO energy gap (in electronvolt) given the molecular graph. We note that the subsetted splits used in this work could be different from those used in Rampášek et al. (2022) as the `numpy` random generator may not be persistent across `numpy` versions[2]. To enable reproducibility, we also make our split indices available for future studies to benchmark against.

**MoleculeNet small datasets** Hu et al. (2020b) (MIT License) We follow Sun et al. (2022) and use the selection of five small molecular property prediction datasets from the MoleculeNet benchmarks, including BBBP, BACE, Tox21, ToxCast, and SIDER. Each graph is a molecule, and it is processed the same way as for MolHIV and MolPCBA. All these datasets adopt the *scaffold splitting* strategy that is similarly used on MolHIV and MolPCBA.

**Peptides-func & Peptides-struct** Dwivedi et al. (2022c) (CC BY-NC 4.0 License) both contain the same 16K peptide graphs retrieved from SAT-Pdb Singh et al. (2016), whose nodes are residues. The two datasets differ in the graph-level tasks associated with them. Peptides-func aims to predict the functions of each peptide (10-way multilabel classification), while Peptides-struct aims to regress 11 structural properties of each peptide. Splitting is done via meta-class holdout based on the original labels of the peptides.

**CIFAR10 & MNIST** Dwivedi et al. (2022a) (CC BY-SA 3.0 and MIT License) are derived from the CIFAR10 and MNIST image classification benchmarks by converting the images into SLIC superpixel graphs with 8 nearest neighbors for each node (superpixel). The 10-class classification and the splitting follow the original benchmarks (MNIST 55K/5K/10K, CIFAR10 45K/5K/10K train/validation/test splits).

**CSL** Dwivedi et al. (2022a) (MIT License) contains 150 graphs that are known as circular skip-link graphs Murphy et al. (2019). The goal is to classify each graph into one of ten isomorphism classes. The dataset is class-balanced, where each isomorphism class contains 15 graph instances. Splitting is done by stratified five-fold cross-validation.

---

[2] https://stackoverflow.com/a/71790820/12519564

**EXP** Abboud et al. (2021) (*unknown* License) contains 600 pairs of graphs (1,200 graphs in total) that cannot be distinguished by 1&2-WL tests. The goal is to classify each graph into one of two isomorphism classes. Splitting is done by stratified five-fold cross-validation.

**arXiv** (ODC-BY License) Hu et al. (2020a) is a directed citation graph whose nodes are arXiv papers and whose edges are citations. Each node is featured by a 128-dimensional embedding obtained by averaging over the word embeddings of the paper's title and abstract. The goal is to classify the papers (nodes) into one of 40 subject areas of arXiv CS papers. Papers published before 2017 are used for training, while the remaining papers that are published before and after 2019 are used for validation and testing.

**Proteins** (CC0 License) Hu et al. (2020a) is an undirected and weighted graph representing the interactions (edges) between proteins (nodes) obtained from eight species. Each edge has eight channels, corresponding to different types of protein interaction evidence. The task is to predict proteins' functions (112-way multilabel classification). The splitting is done by holding out proteins that correspond to specific species.

# E    VISUALIZATION OF GPSE ENCODINGS ON 1-WL INDISTINGUISHABLE GRAPH PAIRS

**Definition 4** (Graph isomorphism). *Two graphs $G$ and $H$ are isomorphic if there exists a bijection $f$ between their vertex sets*

$$f : V(G) \rightarrow V(H)$$

*s.t. any two vertices $u, v \in G$ are adjacent in $G$ if and only if $f(u), f(v) \in H$ are adjacent in $H$.*

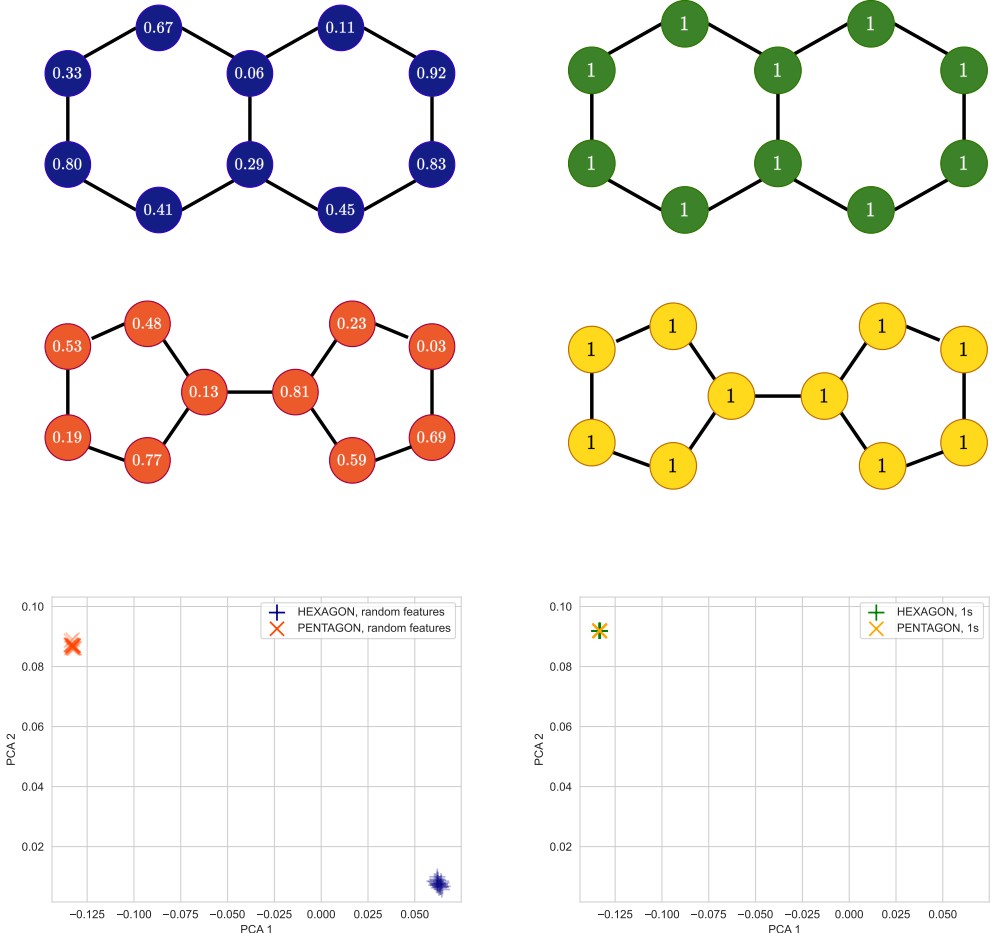

Figure E.1: Visualization of GPSE encodings on 1-WL indistinguishable graph pairs. Applying GPSE with randomly initialized node features results in distinct encodings for HEXAGON (indigo) and PENTAGON (orange) graphs (left). The same graphs cannot be distinguished by our encoder when the node features are set to 1 for each node (right).

The 1-Weisfeiler-Leman (WL) test is an algorithm akin to message-passing that is commonly used to detect non-isomorphic graphs. It can also be viewed as a measure of expressivity: A GNN that can distinguish all pairs of non-isomorphic graphs that can also be distinguished by the 1-WL test is called "1-WL expressive".

In Section 3.3, we discussed that our GPSE is expressive enough to discern graphs that are 1-WL indistinguishable, a well-known limitation of MPNNs Xu et al. (2019). However, Sato et al. (2021) show that the 1-WL expressivity limitation exists only when each node employs identical features; appending random features to the nodes is sufficient to achieve expressivity that goes beyond 1-WL. GPSE leverages precisely this property by replacing the node features by vectors drawn from a standard Normal distribution, such that no two graphs have identical node features.

Here, we demonstrate the importance of these random node features empirically. Consider the following two graphs displayed in Fig. E.1: One resembles two hexagons sharing an edge (referred to as HEXAGON), while the other resembles two pentagons connected by an edge (PENTAGON). These are a well-known pair of non-isomorphic but 1-WL indistinguishable graphs. Non-isomorphism implies that these graphs do not share the same connectivity (see Def. 4 for a formal definition). The WL test also has its limitations: While two graphs that are deemed non-isomorphic are guaranteed to be so, there are cases where it cannot detect non-isomorphism as in the HEXAGON-PENTAGON case.

In our experiment, we create two sets of graphs, both consisting of 20 copies of HEXAGON and PENTAGON graphs each. The two sets are identical except one has all node features set to 1, while the other has features drawn from a random Normal assigned to each node, thus mirroring the actual GPSE training pipeline. We then apply an already trained GPSE encoder (trained on ZINC) to both sets and for each graph we generate aggregated (graph-level) encodings by averaging the obtained 512-dimensional node encodings from GPSE. For visualization purposes, we then apply dimensionality reduction to these graph-level encodings by first fitting a 2-dimensional PCA to GPSE encodings generated on ZINC, and then applying it to the encodings from the synthetic data.

As shown in Fig. E.1, applying GPSE to graphs with randomly initialized node features results in distinct encodings for HEXAGON (indigo) and PENTAGON (orange) graphs (Fig. E.1, left). The same graphs cannot be distinguished by our encoder when the node features are 1 for each node (Fig. E.1, right). The same result is observed when analysing the graph-level PCA embeddings, that can clearly separate the two types of graphs when random node features are used by GPSE, but not otherwise. This underlines the importance of randomized node features in GPSE.

# F    ADDITIONAL RESULTS

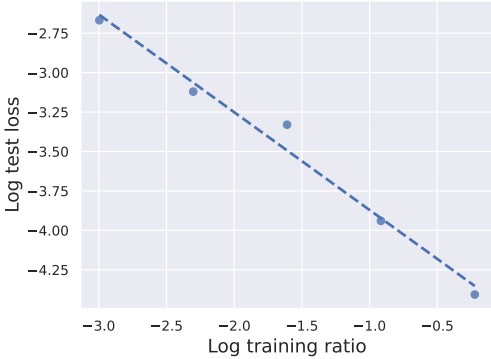

Figure F.1: Training size scaling law for GPSE on MolPCBA.

Table F.1: GPSE architecture ablation. Held-out positional and structural encodings prediction performance when trained on 5% MolPCBA ($R^2$ scores ↑). Ablated settings of the GPSE architecture are listed and compared to the full GPSE settings. The performance of the full GPSE architecture is shown in the bottom row.

| Ablated setting | GPSE default | Overall | ElstaticPE | LapPE | RWSE | HKdiagSE | EigValSE | CycleSE |
|---|---|---|---|---|---|---|---|---|
| 10 layers | 20 layers | 0.9585 | 0.9376 | 0.9302 | 0.9645 | 0.9622 | 0.9543 | 0.9701 |
| 128 dim | 512 dim | 0.9688 | 0.9484 | 0.9501 | 0.9729 | 0.9734 | 0.9706 | 0.9739 |
| GCN | GatedGCN | 0.0409 | 0.0408 | 0.0325 | 0.0424 | 0.0396 | 0.0483 | 0.0410 |
| GIN | GatedGCN | 0.6095 | 0.6953 | 0.4180 | 0.6237 | 0.6349 | 0.4002 | 0.6391 |
| GATv2 | GatedGCN | 0.9580 | 0.9560 | 0.9476 | 0.9643 | 0.9530 | 0.9679 | 0.9561 |
| No VN | VN | 0.9478 | 0.9340 | 0.9359 | 0.9552 | 0.9479 | 0.9314 | 0.9568 |
| Shared MLP head | Indep. MLP heads | 0.9751 | 0.9619 | 0.9644 | 0.9802 | 0.9764 | 0.9714 | **0.9778** |
| GPSE | | **0.9790** | **0.9638** | **0.9725** | **0.9837** | **0.9808** | **0.9818** | 0.9774 |

Table F.2: Four PSE augmentations combined with five different GNN models evaluated on the PCQM4Mv2-subset dataset. Performance is evaluated as mean absolute error (MAE ↓) and averaged over 4 seeds.

| | GCN | GatedGCN | GIN | GINE | Transformer | Avg. reduction |
|---|---|---|---|---|---|---|
| none | 0.1934 ± 0.0012 | 0.1845 ± 0.0031 | 0.1790 ± 0.0011 | 0.1364 ± 0.0011 | 0.4193 ± 0.0167 | – |
| rand | 0.7604 ± 0.0019 | 0.7515 ± 0.0027 | 0.7532 ± 0.0021 | 0.4269 ± 0.0068 | 0.9810 ± 0.0064 | N/A |
| LapPE | 0.1834 ± 0.0023 | 0.1757 ± 0.0010 | 0.1720 ± 0.0018 | 0.1338 ± 0.0006 | 0.2433 ± 0.0056 | 17.76% |
| RWSE | 0.1877 ± 0.0025 | 0.1782 ± 0.0012 | **0.1695 ± 0.0007** | 0.1317 ± 0.0005 | 0.1930 ± 0.0016 | 26.60% |
| GPSE | **0.1822 ± 0.0028** | **0.1715 ± 0.0011** | 0.1713 ± 0.0011 | **0.1294 ± 0.0006** | **0.1909 ± 0.0019** | **28.66%** |

Table F.3: GPSE pre-training ablations

(a) Virtual node, convolution type, and layers ablation using 5% MolPCBA for training.

| Layers | GPSE (GatedGCN) | | GPSE (GIN) | |
|---|---|---|---|---|
| | VN | no VN | VN | no VN |
| 5 | 0.8387 | 0.6982 | 0.4879 | 0.1347 |
| 10 | 0.9585 | 0.8353 | 0.5156 | 0.2476 |
| 15 | 0.9716 | 0.9231 | 0.5887 | 0.2523 |
| 20 | 0.9778 | 0.9478 | 0.6095 | 0.2743 |
| 30 | 0.9806 | 0.9559 | 0.4149 | 0.3740 |
| 40 | 0.9782 | 0.9459 | 0.5420 | 0.3968 |

(b) Training size scaling law for GPSE on MolPCBA.

| Training size | Overall test loss (MAE + cosine loss) ↓ |
|---|---|
| 5% | 0.06939 |
| 10% | 0.04414 |
| 20% | 0.03579 |
| 40% | 0.01945 |
| 80% | 0.01219 |

Table F.4: GPSE training task ablation. The colors indicate whether a particular PSE task for training GPSE improves or worsens the downstream performance.

| Excluded task | PCQM4Mv2 (subset) MAE ↓ | ogbg-molhiv AUROC ↑ |
|---|---|---|
| – | 0.1196 ± 0.0004 | 0.7815 ± 0.0133 |
| LapPE & EigVals | 0.1200 ± 0.0006 | 0.7849 ± 0.0067 |
| ElstaticPE | 0.1197 ± 0.0007 | 0.7681 ± 0.0146 |
| RWSE | 0.1205 ± 0.0006 | 0.7771 ± 0.0105 |
| HKdiagSE | 0.1202 ± 0.0004 | 0.7787 ± 0.0198 |
| CycleSE | 0.1199 ± 0.0011 | 0.7739 ± 0.0240 |

Table F.5: GPSE training dataset ablation. Performance measured in MAE ↓. Green indicates fine-tuning GPSE on specific downstream dataset helps improve the performance. **Bold** indicates the best performance achieved on a particular downstream task.

| Training dataset | # unique graphs | Avg. # nodes | ZINC (subset) | | PCQM4Mv2 (subset) | |
|---|---|---|---|---|---|---|
| | | | Not finetuned | Finetuned | Not finetuned | Finetuned |
| GEOM | 169,925 | 18 | 0.0707 ± 0.0086 | 0.0685 ± 0.0055 | 0.1196 ± 0.0005 | 0.1194 ± 0.0002 |
| ZINC | 219,384 | 23 | 0.0700 ± 0.0041 | – | 0.1202 ± 0.0005 | 0.1197 ± 0.0007 |
| PCQM4Mv2 | 273,920 | 14 | 0.0721 ± 0.0042 | 0.0713 ± 0.0014 | **0.1192 ± 0.0005** | – |
| ChEMBL | 970,963 | 30 | 0.0667 ± 0.0079 | **0.0643 ± 0.0036** | 0.1195 ± 0.0003 | 0.1195 ± 0.0005 |
| MolPCBA | 323,555 | 25 | 0.0648 ± 0.0030 | 0.0668 ± 0.0076 | 0.1196 ± 0.0004 | 0.1195 ± 0.0007 |

Table F.6: Extended results for Table 4: Performance on MoleculeNet small datasets (scaffold test split), evaluated in AUROC (%)↑. Red indicates worse than baseline performance.

| | BBBP | BACE | Tox21 | ToxCast | SIDER | ClinTox | MUV | HIV |
|---|---|---|---|---|---|---|---|---|
| No pre-training (baseline) (Hu et al., 2020b) | 65.8 ± 4.5 | 70.1 ± 5.4 | 74.0 ± 0.8 | 63.4 ± 0.6 | 57.3 ± 1.6 | 58.0 ± 4.4 | 71.8 ± 2.5 | 75.3 ± 1.9 |
| SSL InfoMax pre-trained (Hu et al., 2020b) | 68.8 ± 0.8 | 75.9 ± 1.6 | 75.3 ± 0.5 | 62.7 ± 0.4 | 58.4 ± 0.8 | 69.9 ± 0.3 | 75.3 ± 2.5 | 76.0 ± 0.7 |
| SSL EdgePred pre-trained (Hu et al., 2020b) | 67.3 ± 2.4 | 79.9 ± 0.9 | 76.0 ± 0.6 | 64.1 ± 0.6 | 60.4 ± 0.7 | 64.1 ± 3.7 | 74.1 ± 2.1 | 76.3 ± 1.0 |
| SSL AttrMasking pre-trained (Hu et al., 2020b) | 64.3 ± 2.8 | 79.3 ± 1.6 | 76.7 ± 0.4 | 64.2 ± 0.5 | 61.0 ± 0.7 | 71.8 ± 4.1 | 74.7 ± 1.4 | 77.2 ± 1.1 |
| SSL ContextPred pre-trained (Hu et al., 2020b) | 68.0 ± 2.0 | 79.6 ± 1.2 | 75.7 ± 0.7 | 63.9 ± 0.6 | 60.9 ± 0.6 | 65.9 ± 3.8 | 75.8 ± 1.7 | 77.3 ± 1.0 |
| GraphCL pre-trained (You et al., 2020b) | 69.7 ± 0.7 | 75.4 ± 1.4 | 73.9 ± 0.7 | 62.4 ± 0.6 | 60.5 ± 0.9 | 76.0 ± 2.7 | 69.8 ± 2.7 | 78.5 ± 1.2 |
| InfoGraph pre-trained (Wang et al., 2022a) | 66.3 ± 0.6 | 64.8 ± 0.8 | 68.1 ± 0.6 | 58.4 ± 0.6 | 57.1 ± 0.8 | 66.3 ± 0.6 | 44.3 ± 0.6 | 70.2 ± 0.6 |
| JOAOv2 pre-trained (Wang et al., 2022a) | 66.4 ± 0.9 | 67.4 ± 0.7 | 68.2 ± 0.8 | 57.0 ± 0.5 | 59.1 ± 0.7 | 64.5 ± 0.9 | 47.4 ± 0.8 | 68.4 ± 0.5 |
| GraphMAE (Hou et al., 2022) | 72.0 ± 0.6 | 83.1 ± 0.9 | 75.5 ± 0.6 | 64.1 ± 0.3 | 60.3 ± 1.1 | 82.3 ± 1.2 | 76.3 ± 2.4 | 77.2 ± 1.0 |
| GraphLoG (Xu et al., 2021) | 72.5 ± 0.8 | 83.5 ± 1.2 | 75.7 ± 0.5 | 63.5 ± 0.7 | 61.2 ± 1.1 | 76.7 ± 3.3 | 76.0 ± 1.1 | 77.8 ± 0.8 |
| GraphLoG (Feature extractor) | 65.6 ± 1.0 | 82.5 ± 1.2 | 73.2 ± 0.5 | 63.6 ± 0.4 | 60.9 ± 0.7 | - | - | - |
| LapPE augmented | 67.1 ± 1.6 | 80.4 ± 1.5 | 76.6 ± 0.3 | 65.9 ± 0.7 | 59.3 ± 1.7 | 76.4 ± 2.3 | 75.6 ± 0.8 | 75.6 ± 1.1 |
| RWSE augmented | 67.0 ± 1.4 | 79.6 ± 2.8 | 76.3 ± 0.5 | 65.6 ± 0.3 | 58.5 ± 1.4 | 74.5 ± 4.4 | 75.0 ± 1.0 | 78.1 ± 1.5 |
| AllPSE augmented | 67.6 ± 1.2 | 77.0 ± 4.4 | 75.9 ± 1.0 | 63.9 ± 0.3 | 63.0 ± 0.6 | 72.6 ± 4.3 | 67.9 ± 0.7 | 75.4 ± 1.5 |
| GPSE augmented | 66.2 ± 0.9 | 80.8 ± 3.1 | 77.4 ± 0.8 | 66.3 ± 0.8 | 61.1 ± 1.6 | 78.8 ± 3.8 | 76.6 ± 1.2 | 77.2 ± 1.5 |

Table F.7: Extended results for Table 6: OOD transferability to OGB node classification benchmarks.

|  | +GPSE? | **arXiv** **ACC** (%) ↑ | **Proteins** **AUROC** (%) ↑ |
|---|---|---|---|
| GCN | ✗ | 71.74 ± 0.29 | 79.91 ± 0.24 |
|  | ✓ | 71.67 ± 0.12 | 79.62 ± 0.12 |
| SAGE | ✗ | 71.74 ± 0.29 | 80.35 ± 0.07 |
|  | ✓ | 72.19 ± 0.32 | 80.14 ± 0.22 |
| GAT(E)v2 | ✗ | 71.69 ± 0.21 | 83.47 ± 0.13 |
|  | ✓ | 72.17 ± 0.42 | 83.51 ± 0.11 |
| Transformer | ✗ | 57.00 ± 0.79 | 73.93 ± 1.44 |
|  | ✓ | 59.17 ± 0.21 | 74.67 ± 0.74 |
| GPS | ✗ | 70.60 ± 0.28 | 69.55 ± 5.67 |
|  | ✓ | 70.89 ± 0.36 | 72.05 ± 3.75 |
| *Best of above* | ✗ | 71.74 ± 0.29 | 83.47 ± 0.13 |
|  | ✓ | 72.19 ± 0.32 | 83.51 ± 0.11 |

# G  GPSE TRAINING AND INFERENCE TIMES

Table G.1: GPSE training times. Target PSE pre-computation included.

| Training dataset | Num. unique graphs | Target PSE pre-comp time | Time (epoch/total) | Full training time |
|---|---|---|---|---|
| MolPCBA (default) | 323,555 | 1.58h | 596s / 19.88h | 21.46h |
| PCQM4Mv2-full | 273,920 | 0.87h | 429s / 14.30h | 15.17h |
| ZINC-full | 219,384 | 0.89h | 398s / 13.26h | 14.15h |
| GEOM | 169,925 | 0.78h | 321s / 10.69h | 11.47h |
| ChEMBL | 970,963 | 5.97h | 2509s / 83.65h | 89.62h |

Table G.2: GPSE inference times. LapPE and RWSE computation times are included for comparison. Missing entries are due to experimental settings not included in the benchmarking experiments. (*: Obtained from the GPS paper; **: Neighbor batched computation (batch size: 1024, neighbor sizes: 30, 20, 10, 5, . . . , 5))

| Dataset | Num. graphs | Time (GPSE) | Time (LapPE) | Time (RWSE) |
|---|---|---|---|---|
| ZINC-subset | 12,000 | 6 sec | 25 sec | 11 sec |
| PCQM4Mv2-subset | 446,405 | 3.57 min | 3.88 min | 7.32 min |
| PCQM4Mv2-full | 3,746,620 | 31.15 min | – | 51 min * |
| MolHIV | 41,127 | 23 sec | 37 sec | 58 sec * |
| MolPCBA | 437,929 | 4.6 min | 6.13 min | 8.33 min * |
| Peptides | 15,535 | 28 sec | 73 sec * | – |
| CIFAR10 | 60,000 | 2.15 min | 2.55 min * | – |
| MNIST | 70,000 | 100 sec | 96 sec * | – |
| arXiv | 1 | 4 sec | – | – |
| Proteins | 1 | 6.68 min ** | – | – |

# H GPSE VS. CONVENTIONAL PSE SCALING EXPERIMENTS

Table H.1: Runtimes of each PSE computation with respect to percentage of dataset used.

| PSE / % MolPCBA | 0.1% | 0.3% | 0.5% | 0.8% | 1% | 3% | 5% | 8% | 10% | 25% |
|---|---|---|---|---|---|---|---|---|---|---|
| GPSE | 1s | 1s | 1s | 2s | 3s | 9s | 15s | 24s | 32s | 1m 16s |
| AllPSE | 12s | 46s | 41s | 1m 2s | 1m 15s | 4m 15s | 7m 13s | 12m 3s | 14m 22s | 39m 52s |
| LapPE | 1s | 1s | 3s | 3s | 5s | 15s | 24s | 35s | 55s | 1m 53s |
| RWSE | 1s | 1s | 3s | 3s | 4s | 17s | 25s | 36s | 44s | 1m 41s |
| ElstaticPE | 1s | 1s | 2s | 2s | 3s | 10s | 20s | 33s | 53s | 2m 12s |
| HKdiagSE | 1s | 1s | 2s | 3s | 4s | 11s | 22s | 31s | 48s | 3m 14s |
| CycleGE | 6s | 17s | 28s | 44s | 58s | 2m 57s | 4m 35s | 7m 12s | 9m 15s | 27m 20s |

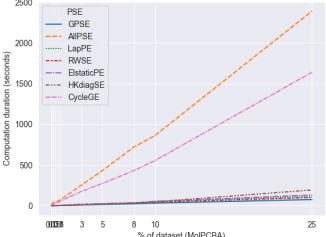

(a) GPSE + individual PSEs + combined PSEs (AllPSE)

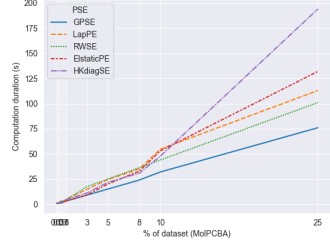

(b) GPSE + individual PSEs only

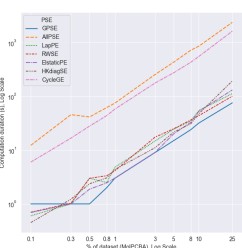

(c) Log-log plot of GPSE + individual PSEs + combined PSEs (AllPSE)

Figure H.1: Scaling of PSE computation time with respect to number of graphs as % of MolPCBA dataset used. Visualization of Table H.1.

Table H.2: Runtimes of each PSE computation with respect to average graph sizes in dataset. CycleGE is excluded both in itself and as part of AllPSE, as cycle counting on large, dense and regular Erdős-Rényi graphs become computationally infeasible.

| PSE / Graph size | 100 | 300 | 500 | 1000 |
|---|---|---|---|---|
| GPSE | 1s | 7s | 27s | 1m 29s |
| AllPSE (No CycleGE) | 8s | 44s | 2m 15s | 11m 20s |
| LapPE | 2s | 9s 250ms | 34s | 2m 35s |
| RWSE | 2s | 9s 760ms | 31s 480ms | 3m 27s |
| ElstaticPE | 1s 500ms | 10s 670ms | 25s 30ms | 2m 19s |
| HKdiagSE | 2s | 13s | 44s | 2m 44s |

In these experiments, we measure and compare the computation time of GPSE with those of individual PSEs used in the pre-training of the GPSE model, as well as their combination (AllPSE). We conducted two sets of experiments. In the first, we used a dataset of similarly sized graphs in MolPCBA, but ran PSE computation for an increasing percentage of the dataset (Figure H.1). In the second, we generated multiple datasets of 1000 synthetic (Erdős-Rényi) graphs, scaling up the number of nodes per graph (Figure H.2) in each.

In both sets of experiments, GPSE is considerably faster to compute than the individual PSEs, and orders-of-magnitude faster than computing and combining all PSEs as AllPSE. Additionally, we observed that GPSE scales better than individual and combined PSEs. The better scaling properties of GPSE are particularly evident in scaling graph sizes (Figure H.2): As we scaled up the number of nodes in a graph to 1000, the advantage of GPSE became more apparent. This is somewhat an expected result: Regardless of graph size, inference of GPSE involves $O(Lm)$ computations, where L is the number of layers, and m is the number of edges; thus GPSE scales linearly on the number of edges in a graph. On the other hand, LapPE, for example, is expected to have polynomial complexity, as it involves eigendecomposing the graph Laplacian.

Another important point to highlight is that at inference time, *the complexity of GPSE remains unchanged* regardless of how many types of PSEs were used to train the model. This leads to

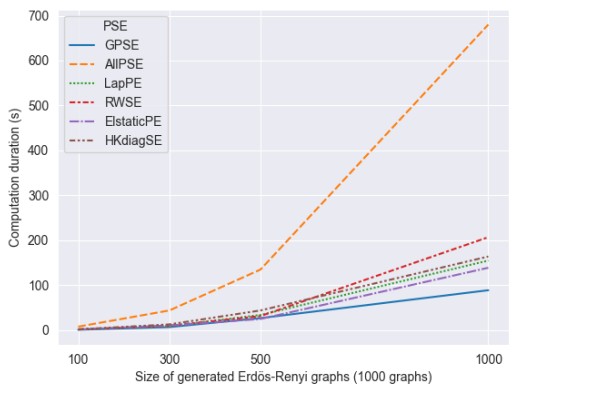 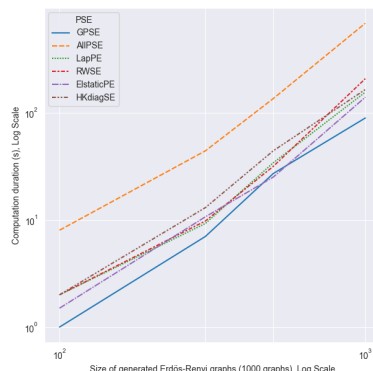

(a) GPSE + individual PSEs + combined PSEs (AllPSE)

(b) Log-log plot of GPSE + individual PSEs + combined PSEs (AllPSE)

Figure H.2: Scaling experiments with respect to size of graphs, keeping the number of graphs in each dataset constant. Visualization of Table H.2.

significant advantages over AllPSE, which relies on computing and concatenating all PSEs. The scalability issues of AllPSE is additionally exacerbated when useful but highly expensive PSEs such as CycleGE are used.

