# OpenReview forum: "Graph Positional and Structural Encoder"
_ICLR.cc/2024/Conference — Submitted to ICLR 2024_

### Official Review · Reviewer_rYLm · 2023-10-30

**Soundness:** 2 fair
**Presentation:** 4 excellent
**Contribution:** 2 fair
**Rating:** 6
**Confidence:** 4

**Summary:**

This paper introduces the graph positional and structural encoder (GPSE), which is the first attempt to create a graph encoder that can generate rich PSE representations for GNNs. GPSE can learn representations for a variety of PSEs and is notably adaptable. When trained on a large molecular graph dataset, it can be applied successfully to different datasets, even those with differing distributions or modalities. The study shows that GPSE-enhanced models can either improve performance or match the results of models using explicitly computed PSEs. This contribution suggests the potential for large pre-trained models in extracting graph positional and structural information, positioning them as strong alternatives to current methods.

**Strengths:**

- The paper is well structured, easy to follow.
- The introduced GPSE model stands out for its simplicity, consistently delivering results either superior to or on par with hand-crafted PSEs.
- A commendable effort has been made in terms of the ablation studies, providing comprehensive insights into the proposed model.

**Weaknesses:**

1. **Comparison with related work:** While the paper does touch upon the LSPE (Dwivedi et al. 2021) – which shares similarities in learning positional representations along with the prediction task – a more detailed comparison is crucial to identify the unique contributions of the present work.
2. **Clarity on computational complexity:** A major limitation of most hand-crafted PSEs is their high complexity on large graphs. The paper could benefit from a deeper dive into GPSE's computational complexity, particularly when compared against the often complex hand-crafted PSEs and other encoding strategies.
3. **Missing baseline method:** Throughout the experiments, GPSE is compared against individual PSEs. Considering GPSE is designed to capture multiple PSE representations, a comparison against a combined baseline which concatenates various PSEs – akin to those used in GPSE's pre-training – would offer a more fair evaluation.

**Minor points**
- **Assumptions about Audience:** The introduction might be challenging for those new to the field, given the use of specific terms like "1-WL bounded expressiveness" and "over-squashing" without much elucidation.

_Reference:_

Dwivedi, Vijay Prakash, et al. "Graph Neural Networks with Learnable Structural and Positional Representations." ICLR 2021.

**Questions:**

- Q1: For downstream tasks, is the pretrained GPSE kept frozen or is it finetuned? Would fine-tuning the GPSE improve its performance? It is noteworthy that, in other modalities like NLP and computer vision, fine-tuning pretrained models often outperforms freezing pretrained models.

I will be happy to increase my rating if the authors can address all my concerns and questions.

---

> ### Author Response · Authors · 2023-11-18
> **Reply to Reviewer rYLm**
>
> > Comparison with related work: While the paper does touch upon the LSPE (Dwivedi et al. 2021) – which shares similarities in learning positional representations along with the prediction task – a more detailed comparison is crucial to identify the unique contributions of the present work.
>
> We thank the reviewer for raising the question. We kindly refer the reviewer to our global response, “Relation between GPSE and LPSE,” for a more in-depth explanation of the difference between the two.
>
> > Clarity on computational complexity: A major limitation of most hand-crafted PSEs is their high complexity on large graphs. The paper could benefit from a deeper dive into GPSE's computational complexity, particularly when compared against the often complex hand-crafted PSEs and other encoding strategies.
>
> We appreciate the reviewer’s constructive comment regarding the computational complexity of GPSE. As the reviewer has suggested, we added a detailed analysis with an empirical study demonstrating the computational benefit of using GPSE instead of directly computing hand-crafted PSEs. As a concrete example, computing all PSEs on MolPCBA took about four hours, while GPSE only took about four minutes. We kindly refer the reviewer to the global response, “Computational complexity of GPSE,” for more details, as well as Appendix [H] for results and additional visualizations.
>
> > Missing baseline method: Throughout the experiments, GPSE is compared against individual PSEs. Considering GPSE is designed to capture multiple PSE representations, a comparison against a combined baseline which concatenates various PSEs – akin to those used in GPSE's pre-training – would offer a more fair evaluation.
>
> We thank the reviewer for bringing up this valid point. We kindly refer the reviewer to our global response, “Comprehensive comparison against AllPSE baselines,” for more detailed information about the newly added baseline as suggested. In brief, we observed that GPSE outperforms or matches the performance of AllPSE in nearly all of our primary benchmarks.
>
> > Assumptions about Audience: The introduction might be challenging for those new to the field, given the use of specific terms like "1-WL bounded expressiveness" and "over-squashing" without much elucidation.
>
>
> The WL-test is explained along with visual support in Appendix E, in discussion of how GPSE breaks symmetries associated with the test. Over-squashing and over-smoothing are similarly explained in Appendix C: Theory details. We do agree with the reviewer, however, that connections to these sections should be clarified to redirect the reader. To this end, we have re-adjusted the beginning of Section 2.2 to refer to Appendix C as well as C.1, where we further discuss the connections between over-smoothing/over-squashing and GPSE. We have similarly referred to Appendix E in the beginning of Section 3.3, and reformatted Appendix E to further elucidate the WL test and its implications.
>
> > Q1: For downstream tasks, is the pretrained GPSE kept frozen or is it finetuned? Would fine-tuning the GPSE improve its performance? It is noteworthy that, in other modalities like NLP and computer vision, fine-tuning pretrained models often outperforms freezing pretrained models.
>
> We thank the reviewer for the insightful comments. We note that all GPSE results presented in the main paper are obtained with a GPSE model with frozen weights (no fine-tuning). On the flip side, we experimented with fine-tuning but no concrete conclusion about whether it is helpful was drawn. Specifically, we have tried fine-tuning GPSE on the two primary molecular property benchmarking tasks: ZINC and PCQM4Mv2. As we have presented in Appendix Table F.5, fine-tuning could help in certain cases, but no consistent pattern shows the superiority of fine-tuning.
>
> Fine-tuning pre-trained models is a vast topic, and how to effectively do so for better downstream performance in our specific setting is an under-explored area. Thus, an exciting future avenue is to study the optimal way to fine-tune GPSE to further adapt to specific downstream datasets.
>
> ---
> We sincerely appreciate the reviewer's constructive feedback and have diligently addressed each concern raised in our revision. We remain open to further discussion and are ready to address any additional concerns the reviewer may have. In light of these comprehensive revisions, we kindly ask the reviewer to reevaluate our work, hoping that the enhancements made positively impact the overall quality and merit of our paper.

---

> > ### Comment · Reviewer_rYLm · 2023-11-22
> > **Response to the rebuttal**
> >
> > Thank you for your detailed response, which has addressed most of my concerns. Therefore, I am leaning towards acceptance.

---

> > > ### Author Response · Authors · 2023-11-22
> > > **Reply to Reviewer rYLm**
> > >
> > > Thank you, reviewer rYLm, for acknowledging our efforts in addressing the concerns raised in your review. We are pleased to hear that our rebuttal has met your expectations. If there are any remaining questions or issues, please feel free to bring them to our attention. We are eager to engage in further discussion to clarify any points and ensure the quality of our work meets the highest standards.

---

### Official Review · Reviewer_7nJh · 2023-10-30

**Soundness:** 1 poor
**Presentation:** 2 fair
**Contribution:** 1 poor
**Rating:** 5
**Confidence:** 4

**Summary:**

The paper proposes GPSE, a framework that can be used to pretrain an encoder in order to _learn_ a diverse set of positional and structural encodings, given an input graph augmented with random node features. The authors show good empirical results on a diverse set of downstream tasks when using the pretrained encoder to extract the encodings.

**Strengths:**

Results on some datasets are promising, like those obtained on ZINC.

**Weaknesses:**

The authors have __not__ shown the advantage of learning the encodings instead of computing them directly on the dataset of downstream task of interest. In each table the direct competitor should be the GNN predictor directly augmented with __all__ the positional and structural encodings (that are learned by GPSE). For example, Table 2 should include "GPS + LapPE + ElstaticPE + EigValSE + RSWE + HKdiagSE + CycleSE". Comparing only to "GPS + LapPE" or "GPS + RWSE" is unfair, and no conclusion can be drawn from it as it might be that simply using one of the other encodings lead to better results, so the role of the pretraining is less clear.

It is unclear how the authors deal with sign and basis ambiguity. Consider for example LapPE. When training GPSE, what is the sign of the target LapPE? What about the basis?

**Questions:**

Please clarify the importance of learning the encodings. Otherwise the conclusion is that using all those encodings is better than using no encoding, and and it is unclear what is the impact of computing them directly on the input graph instead of learning them in a pretraining step that uses MolPCBA.
Also consider the problem of sign and basis ambiguity.

---

> ### Comment · Reviewer_vPPW · 2023-11-16
>
> Regardless of the anonymity issue that I raised, I would also like to discuss the point made by Reviewer 7nJh and Reviewer rYLm, since positional and structural encodings are significant topics in recent graph learning.
>
> I have a same concern with Reviewer 7nJh and Reviewer rYLm about the fair comparison on Table 2 as noted in my review:
> >For fair comparison in Table 2 and Table 3, GPSE should be compared not only with single combinations like GPS+LapPE or GPS+RWSE, but also with a more comprehensive set-up, including GPS+LapPE+ElstaticPE+RWSE+EigValSE+HKdiagSE+CycleSE, with simple projection layers for each PSE type.
>
> Additionally, to provide a detailed analysis, it would be beneficial to include the experimental results in Appendix for each combination of encodings (such as GPS+LapPE+ElasticPE, GPS+LapPE+RWSE, GPS+RWSE+CycleSE, ...).
>
> I hope that the comments have been helpful in clarifying the contribution of this paper, and I look forward to the authors' response.

---

> ### Author Response · Authors · 2023-11-18
> **Reply to Reviewer 7nJh**
>
> > Table 2 should include "GPS + LapPE + ElstaticPE + EigValSE + RSWE + HKdiagSE + CycleSE". Comparing only to "GPS + LapPE" or "GPS + RWSE" is unfair, and no conclusion can be drawn from it as it might be that simply using one of the other encodings lead to better results, so the role of the pretraining is less clear.
>
> We appreciate the reviewer for bringing up this valid concern regarding a more proper baseline of concatenating all PSEs. We kindly refer the reviewer to our global response, “Comprehensive comparison against AllPSE baselines,” for more detailed information about the newly added baseline as suggested. In brief, we observed that GPSE outperforms or matches the performance of AllPSE in nearly all of our primary benchmarks.
>
> > It is unclear how the authors deal with sign and basis ambiguity. Consider for example LapPE. When training GPSE, what is the sign of the target LapPE? What about the basis?
>
> We thank the reviewer for the question regarding the sign and basis ambiguity issues [1] for LapPE. We have described our approach in Appendix A (Positional and structural encoding tasks detail). To summarize, we tackled the sign ambiguity issue by taking the **absolute values** of the eigenvectors. On the other hand, we did not explicitly resolve the basis ambiguity issue. However, since we only use the eigenvectors associated with the *first four smallest non-trivial eigenvalues*, there is a significantly lower chance of encountering degenerate eigenvalues (the root cause of basis ambiguity) compared to learning against the full set of eigenvectors.
>
> We further note that *current known strategies from the literature for tackling the sign and basis ambiguity do not apply to our problem setting*. In [1], the goal was to utilize the eigenvectors in a way that is sign and basis invariant. In our problem setting, however, we aim to *learn against the PSEs*, thus requiring an **objective function** designed to be sign and basis invariant. Both invariances are invalid for standard regression objective functions such as MSE, MAE, and Cosine Similarity losses. Although we acknowledge the value of tackling this specific problem in a future project, this is ultimately out of this project's scope, as we primarily aim to demonstrate the possibility of learning a compressed representation that captures multiple PSEs.
>
> [1] Lim, Derek, et al. "Sign and Basis Invariant Networks for Spectral Graph Representation Learning." The Eleventh International Conference on Learning Representations. 2022.
>
> > Please clarify the importance of learning the encodings. Otherwise the conclusion is that using all those encodings is better than using no encoding, and and it is unclear what is the impact of computing them directly on the input graph instead of learning them in a pretraining step that uses MolPCBA. Also consider the problem of sign and basis ambiguity.
>
> We thank the reviewer for the insightful suggestions. Please refer to our earlier reply for more information regarding the **performance benefit** of using GPSE compared to the AllPSE counterparts.
>
> Furthermore, GPSE demonstrates a notable **computational benefit**, with significantly shorter computation time on large datasets like MolPCBA than computing all PSEs exactly. Please refer to the global response, “Computational complexity of GPSE,” for a more in-depth discussion, as well as Appendix [H] for results and additional visualizations.
>
> Finally, as discussed in the reply above, we have dealt with the sign ambiguity of LapPE by taking the absolute values. Meanwhile, basis ambiguity remains an open problem due to the unique challenge of designing a general-purpose objective function that is basis invariant.
>
> ---
> We sincerely appreciate the reviewer's constructive feedback regarding the AllPSE baseline and sign/basis ambiguity concern. We believe that our replies have sufficiently addressed the concerns raised. We remain open to further discussion and are ready to address any additional concerns the reviewer may have. In light of these comprehensive revisions, we kindly ask the reviewer to reconsider their initial evaluation and increase their score accordingly if no lingering questions remain.

---

> ### Author Response · Authors · 2023-11-22
> **Friendly Reminder to Reviewer 7nJh for Open Discussion**
>
> Dear Reviewer 7nJh,
>
> We sincerely appreciate your dedication to reviewing our work and the valuable insights provided. In response to your comments, we have thoroughly addressed each concern in our rebuttal. As the deadline for open discussion is quickly approaching, we kindly remind you to review our efforts and share any further feedback or questions you may have. Your additional input is crucial for us to refine our work effectively. We eagerly await your response and thank you again for your invaluable contributions to this process.
>
> Best regards,
>
> The Authors

---

> > ### Comment · Reviewer_7nJh · 2023-11-22
> >
> > Thank you for the time spent in answering my questions.
> >
> > I have particularly appreciated the additional baseline, which was necessary to assess the importance of learning the encodings. For future work, I would recommend adding baseline experiments with all possible combinations of these encodings for a comprehensive comparison.
> >
> > Regarding the sign and basis ambiguity, I believe the answer does not address my concerns. While I can understand leaving the basis ambiguity for future work, I believe that addressing the sign problem in a more principle manner would significantly strengthen the paper. It is well known that taking the absolute value seriously degrades the expressive power of the encodings [1] so I would encourage the authors to think of alternatives. For example, maybe you could learn a sign-invariant function of the eigenvectors instead of the (absolute values of the) eigenvectors, in the spirit of the literature tackling sign and basis ambiguity.
> >
> > I will increase my score to 5. The weaknesses related to sign and basis ambiguity, and the intrinsic hand-crafted nature of the paper prevents me to fully support this work. I think the usage of: (1) residual connections, (2) gating mechanisms (3) virtual nodes make the paper a bit too much feature-engineered. The justification of these choices is intuitive but certainly not formal.
> >
> >
> > [1] Dwivedi et al., 2020. Benchmarking Graph Neural Networks

---

> ### Author Response · Authors · 2023-11-23
> **Reply to Reviewer 7nJh**
>
> Thank you for your response and for acknowledging our efforts in incorporating the new AllPSE baseline into our study.
>
> We concur with your suggestion that additional combinations of PSE could enrich our results. In line with this, we have already incorporated the most significant binary combination, LapPE+RWSE, into our global replies. We are **actively working on integrating more combinations** and will ensure these results are included in the camera-ready version of our paper.
>
> Regarding the remaining concern about sign ambiguity, we understand that our original experiments have not satisfactorily resolved the issue. We have therefore conducted a new set of experiments to demonstrate this aspect further. Particularly, we pretrained four different versions of GPSE using 5% MolPCBA subset (due to time constraints): (1) GPSE-abs takes absolute value of the LapPE (default setting in our paper), (2) GPSE-noabs do not take absolute value of the LapPE, (3) GPSE-signinvar uses a sign invariant loss function for LapPE by taking the minimum of the losses from both signs, (4) GPSE-SignNet uses a randomly initialized SignNet [1] model to generate sign invariant features as the training target for GPSE. We tested the four variations on two molecular (ZINC-subset, PCQM4Mv2-subset) and two transfer benchmarks (Peptides-func, Peptides-struc). Our results indicate that using the default absolute handling of LapPE results in similar or better performance than other handlings, indicating the **effectiveness of using the absolute LapPE for training GPSE**. Given that it is possible that not using absolute LapPE results in better performance than default GPSE, such as in Peptides-func, we acknowledge the value of further exploring optimal ways to handle LapPE sign ambiguities in future work. That said, our results here proved that the default handling using absolute value is sufficient.
>
> Lastly, we respectfully differ from the viewpoint that our methodological choices lack formal justification as suggested by the reviewer. We would like to indicate that the overall reasons that drive most architectural decisions behind GPSE are fairly straight-forward: (1) Learning PSEs that can encode long-distance relationships require deep GNNs to avoid under-reaching, (2) Deep GNNs invariably suffer from over-smoothing and over-squashing. Our proposed architecture is composed of well-established tools in tackling these issues; we additionally provide  **detailed formal proofs in Appendix C** of our paper, providing theoretical underpinnings for these choices. Therefore, we believe our architectural choices are well-justified from both the perspective of theory and ties to prior work on graph bottlenecks, as well as empirically, as the robustness of our results suggests.
>
> We appreciate your valuable feedback and hope our additional justifications have addressed your concerns further. Considering the additional experiments and clarifications provided, we kindly ask the reviewer to reevaluate the score of our submission.
>
> |            | **ZINC (subset)** | **PCQM4Mv2 (subset)** | **Peptides-struct** | **Peptides-func** |
> | :------- | :-------: | :-------: | :-------: | :-------: |
> |            | **MAE ↓** |  **MAE ↓** |  **MAE ↓** | **AP ↑** |
> | GPSE-abs (default) | **0.0957 ± 0.0044** | **0.1216 ± 0.0002** | **0.2516 ± 0.0018** | 0.6584 ± 0.0042 |
> | GPSE-noabs           | 0.1051 ± 0.0046 | 0.1229 ± 0.0006 | 0.2554 ± 0.0025 | **0.6687 ± 0.0119** |
> | GPSE-signinvar      | 0.1116 ± 0.0072 | 0.1243 ± 0.0004 | 0.2594 ± 0.0019 | 0.6619 ± 0.0097 |
> | GPSE-SignNet        | 0.1035 ± 0.0052 | 0.1232 ± 0.0006 | 0.2568 ± 0.0020 | Not run yet |
>
> [1] Lim, Derek, et al. "Sign and Basis Invariant Networks for Spectral Graph Representation Learning." The Eleventh International Conference on Learning Representations. 2022.

---

### Official Review · Reviewer_vPPW · 2023-11-01

**Soundness:** 2 fair
**Presentation:** 2 fair
**Contribution:** 2 fair
**Rating:** 1
**Confidence:** 4

**Summary:**

The paper introduces the Graph Positional and Structural Encoder (GPSE), an approach for training graph encoders to effectively capture positional and structural encodings (PSEs) within graphs.
These encodings are pivotal for identifying node roles and relationships, a challenge accentuated by the inherent absence of a standard node ordering in graph structures.
GPSE shows the capability to learn and apply these PSEs across a wide range of graph datasets, with potential to match or even surpass the performance of explicitly computed PSEs.
The encoder stands out for its adaptability and proficiency in various graph dimensions and types.


The paper details a method to train a Message Passing Neural Network (MPNN) as a graph encoder, focusing primarily on the extraction of positional and structural representations from query graphs based solely on their structure.
This extraction utilizes a set of PSEs in a predictive learning framework.
Post training, the encoder can adeptly apply these PSEs to augment models (both MPNNs and Transformers) across various datasets.
The design includes diverse PSE types such as Laplacian eigenvectors and eigenvalues, electrostatic and random walk structural encodings, heat kernel structural encodings, and cycle counting graph encodings.
These PSEs provide insights into a node's relative position (positional encodings) and the local connectivity patterns (structural encodings), enhancing the understanding of graph structures.

**Strengths:**

This study presents a noteworthy contribution to the field of graph neural networks, particularly in its architectural and methodological approach.
The originality of the paper lies in its integration of PSEs within GNNs, targeting the specific challenges of graph representation learning.
While the concept might not completely redefine the foundational theories of graph neural networks, it innovatively combines existing ideas to enhance the representation and processing of graph structures, demonstrating a balanced level of originality.

In terms of quality and clarity, the paper is methodically sound, though it might benefit from a more in-depth exploration of its theoretical underpinnings.
The clarity of presentation is commendable, with technical details and concepts explained in a manner that strikes a balance between depth and accessibility.
The significance of the research, while notable, appears more confined to immediate practical applications rather than setting a new paradigm. Nevertheless, its potential impact in improving model performance in various graph-related tasks.

**Weaknesses:**

I've encountered a potential issue concerning the anonymity of the authors in the submission, specifically related to the URL (specifically, the owner of Google Drive is not anonymized) in the code that the authors submitted, which might violate the double-blind review process. While I cannot confirm if the names found in the URL indeed belong to the authors, I wanted to raise this as a potential concern for the integrity of the review process. This concern is based on the `Anonymous Url` at the top of this page and the `Source code submission` section in the Author Guide.

---

The research primarily excels in engineering advancements for graph neural networks, focusing more on practical architectural solutions than on extending theoretical foundations.
Although these innovations address key issues in GNNs, they potentially add complexity and computational overhead that aren't thoroughly examined.

The model's performance, when combined with GPSE, is less impressive, considering its complexity, which includes 20 MPNN layers, MLP heads for each PSE, a gating mechanism, and a virtual node.

For fair comparison in Table 2 and Table 3, GPSE should be compared not only with single combinations like GPS+LapPE or GPS+RWSE, but also with a more comprehensive set-up, including GPS+LapPE+ElstaticPE+RWSE+EigValSE+HKdiagSE+CycleSE, with simple projection layers for each PSE type.

In Table 4, GPSE appears to be less advantageous compared to the SSL `Self-supervised pre-trained (Hu et al., 2020b)` model.
(Hu et al. outperform GPSE on 3 out of 5 datasets by some margins)

**Questions:**

Could you please verify whether the owner of the Google Drive link is the author?
(if you want to check the name that I found, then I will leave the name in the comment)

---

> ### Author Response · Authors · 2023-11-18
> **Reply to reviewer vPPW (1/2)**
>
> > The research primarily excels in engineering advancements for graph neural networks, focusing more on practical architectural solutions than on extending theoretical foundations. Although these innovations address key issues in GNNs, they potentially add complexity and computational overhead that aren't thoroughly examined.
>
> We thank the reviewer for recognizing the engineering advancements of GPSE and its practical contributions to the community. We understand the importance of providing a more thorough analysis of GPSE’s computational complexity to improve its practical impact further. We kindly refer the reviewer to our global response, “Computational complexity of GPSE,” for more detailed information regarding the newly added analysis. In short, we have demonstrated and explained the computational benefit of using GPSE instead of computing all exact PSEs.
>
> > The model's performance, when combined with GPSE, is less impressive, considering its complexity, which includes 20 MPNN layers, MLP heads for each PSE, a gating mechanism, and a virtual node.
>
> To avoid any potential confusion, we would like to underline that these listed components are *of the GPSE model* that are used to generate latent PSEs *in inference mode*, and do not pertain to the downstream models (GPS/GIN/GCN etc.) used. All GPS experiments have been run with 3 to 10 layers, while GCN and GraphSAGE for the node-level tasks use only 3 message-passing layers, for example. The MoleculeNet experiments (Table 4) are similarly conducted with a 5-layer GINE, identical to Sun et al. (2022). In short, the models we use to obtain the results are appropriate for their respective benchmarking settings.
>
> There is of course a limitation of how much PSEs by themselves can help improve performance, which is bound by how much positional information is useful for the given dataset and task at hand. This also depends on the PSEs that are learned; with more powerful PSEs, we can expect the benefits of GPSE also compound. For the scope of our paper, we would like to emphasize that **we consistently match or beat individual PSEs as well as their combinations (please note the added AllPSE experiments)**, an important result by itself.
>
> Finally, despite the suggested complexities, **GPSE is nevertheless much more efficient than the alternative of explicitly computing PSEs**. We have conducted extensive studies on computational runtimes on both (a) increasing number of graphs, and (b) increasing graph sizes while keeping the number of graphs constant, and demonstrated that GPSE has both better wall-clock times and scaling properties on average than individual PSEs, and their combinations. We kindly refer the reviewer to our global response “Computational complexity of GPSE” as well as Appendix [H] for results and additional visualizations.
>
> > For fair comparison in Table 2 and Table 3, GPSE should be compared not only with single combinations like GPS+LapPE or GPS+RWSE, but also with a more comprehensive set-up, including GPS+LapPE+ElstaticPE+RWSE+EigValSE+HKdiagSE+CycleSE, with simple projection layers for each PSE type.
>
> We appreciate the reviewer for bringing up this valid concern regarding a more proper baseline of concatenating all PSEs. We kindly refer the reviewer to our global response, “Comprehensive comparison against AllPSE baselines”, for more detailed information about the newly added baseline as suggested. In brief, we observed that GPSE outperforms or matches the performance of AllPSE in nearly all of our primary benchmarks.

---

> ### Author Response · Authors · 2023-11-18
> **Reply to reviewer vPPW (2/2)**
>
> > In Table 4, GPSE appears to be less advantageous compared to the SSL Self-supervised pre-trained (Hu et al., 2020b) model. (Hu et al. outperform GPSE on 4 out of 5 datasets by some margins)
>
> We first highlight that the Hu et al. results in fact report the *best of 5 distinct self-supervised pre-trained methods*; we opted to show only the best for each dataset for compactness. Our original impression was that the table became overcrowded and detracted from the main points of our results regarding information transfer to downstream tasks. We thank the reviewer for pointing this out, as the current setup may be misleading in that it puts the SSL methods in Hu et al., 2020b in a better light than they actually are. We have therefore added the expanded version of Table 4 to Appendix [I].
>
> Additionally, we have updated our GPSE results in Table 4 after conducting more hyperparameter search on the *mixing ratio between GPSE and the original graph features* - a standard parameter to tune in GPS. Previously, we did not optimize the GPSE performance on the MoleculeNet benchmarks as we did not intend to imply a direct comparison between SSL and GPSE: **The two approaches are complementary** to each other and the key conclusion we wanted to draw was that **GPSE also shows strong ability to transfer knowledge** from large pre-train datasets to the downstream task. While we still think that was the primary intention, we also understand the reviewer’s concern about GPSE appearing to be less advantageous to SSL on paper. With these latest results, *GPSE matches or outperforms the combined SSL on 7/8 MoleculeNet datasets*, and overall are consistently more powerful than any single SSL method.
>
> Finally, we want to reiterate the difference in GPSE’s and SSL’s abilities to transfer knowledge. SSL methods are constrained to transfer knowledge between graphs that have the exact format: node features and edge features. This is due to the nature of SSL expecting the same input graph format between pre-train and downstream datasets. On the flip side, GPSE by design, is more flexible: it can transfer knowledge to *any* graph structured data, as it only uses graph connectivity for the computation, not node/edge features. This flexibility makes the obtained competitive results against SSL methods even more impressive.
>
> > Could you please verify whether the owner of the Google Drive link is the author? (if you want to check the name that I found, then I will leave the name in the comment)
>
> We greatly appreciate the reviewer's suggestions and good intentions regarding using anonymized cloud storage solutions to anonymize our work for review more strictly. We acknowledge the concern about the ownership of the Google Drive link provided in our submission, as discussed in a main thread. That said, we commit to strictly adhering to the recommended practices of using anonymized cloud storage solutions in all our future submissions. We assure the reviewer that maintaining the integrity of the double-blind review process is of utmost importance to us, and any lapse in anonymity was entirely unintentional. We hope this addresses the reviewer's concern and appreciate their understanding in this matter.
>
> ---
> We sincerely thank the reviewer for their thoughtful and insightful comments, which we have found immensely helpful in enhancing the soundness of our work. In response, we have diligently addressed all the points raised, ensuring that each concern has been thoroughly considered and acted upon. Nevertheless, we remain open to further discussion and are fully prepared to conduct additional experiments if required to refine our work further. Given our comprehensive response to the feedback provided, we kindly request the reviewer to reevaluate our manuscript, hoping that these enhancements will positively impact our paper's overall quality and merit.

---

> ### Comment · Reviewer_vPPW · 2023-11-21
>
> Thank you for the detailed responses from the authors.
>
> I recognize the importance of this research topic, which led me to carefully read the authors' rebuttal as well as the comments from other reviewers.
>
> Regarding the comparison with SSL, I fully understand the authors' response.
>
> For other aspects such as complexity and efficiency, the additional information provided in the Global Replies about inference time suggests that GPSE is efficient.
>
> Yet, when considering the training time, it still raises questions about its practicality (since graph sizes in experiments are mostly small, the comparison based on graph size was evaluated more from a scalability perspective than practicality or efficiency).
>
> With these factors in mind, I would like to discuss further with other reviewers whether the complex computations of GPSE are justifiable.
>
> For the analysis, it would be beneficial to conduct comparative analyses with various combinations beyond GPS+AllPSE (I am aware of the time constraints of this review process, so this suggestion is intended for future improvements).
>
> This is essential since the paper seems to hold more value in engineering aspects rather than theoretical ones, and a detailed approach would provide the readers with rich information.
>
> *(Regarding the issue of anonymity, I would like to continue the discussion with AC.)*

---

> > ### Author Response · Authors · 2023-11-21
> > **Reply to Reviewer vPPW**
> >
> > Thank you, reviewer vPPW, for replying to our rebuttals and recognizing our efforts.
> >
> > We want to answer the complexity-related concern further. First, GPSE can be readily used by the users for downstream tasks without re-training. This is the primary motivation behind our emphasis on GPSE as a plug-and-play model and making our pre-trained weights available for immediate use: Inference time is the primary factor in the usability of GPSE, which we have demonstrated in the global replies, and users do not need to worry about any additional training or fine-tuning. Secondly, training GPSE is still faster overall than conventional pre-training: The training time for GPSE was less than 22 hours on MolPCBA, as we reported in Table G.1. In comparison, the pre-training time for a typical SSL method is usually more than a day. For example, in Hu et al., 2020b, the full pre-training time for molecular tasks is about 35 hours. Thus, we argue that GPSE is no more complex than a typical method that requires pre-training in the molecular deep learning domain.
> >
> > Regarding the more comprehensive evaluation with different combinations of PSEs, we acknowledge the usefulness of providing them to strengthen our paper and thank the reviewer for recognizing the difficulties in conducting them all, given the time constraints. Nevertheless, we continue to include results from more PSE combinations and have updated the global replies with the most crucial binary combination: LapPE+RWSE. We assure the reviewer that we will continue these experiments and update them for the camera-ready version of the paper.

---

### Official Review · Reviewer_6S8q · 2023-11-01

**Soundness:** 3 good
**Presentation:** 1 poor
**Contribution:** 2 fair
**Rating:** 5
**Confidence:** 5

**Summary:**

The paper introduces GPSE a method to extract position and structural encodings (pse) from graphs. It can also be applied to other graph modelling tasks without the explicit need to compute the pse. The methos adopts a self-supervision style to train GPSE comprised of an encoder module which extracts the features and then passes through the decoder (MLP) to obtain the final embeddings. The loss is calculated against each of the six pse which is the target. For training, they use the MolPCBA dataset and test it against different molecular and node classification datasets.

**Strengths:**

- The problem of solving for transferable position and structural encoding is of high importance and relevance in the graph machine learning area.
- The paper shows extensive experiments on a variety of different datasets to validate the method of transferability across different graph types.

**Weaknesses:**

- The GPSE architecture is unclear in section 2.2, even after re-reading the appendix and the section multiple times. It would have been easier to follow if the approach was explained in a step-by-step manner and aided with equations.
- The contributions in the paper seem to be limited as many of the methods have been adopted similarly but is unclear how are they different from the existing approach. (please refer to questions)

**Questions:**

- Section 2.2 talks more about the general effect of over-smoothing and squashing in graph neural networks, but it is not clear what this has to do with the method of learning effective position encoding for graphs. Why is it relevant to the problem?
- In Table 4 GPSE shows better performance on MoleculeNet dataset where it is mentioned that GPSE is at a comparative disadvantage but is this due to the nature of the dataset or the importance of structural and position to the specific dataset? It will be interesting to observe what will be the effect of GINE+{LapPE/RWSE} vs GPSE.
- Why are results from other datasets like (ClinTox MUV HIV) not included in Table 4 as GraphLoG is evaluated on these datasets?
- Why are GNN models used as a baseline for testing the extreme OOD node-classification benchmarks (table 6) instead it will be interesting to see how transformer+GPSE will perform on the dataset as they don't have the inductive bias of GNNs.
- The paper [1] has a similar research question and method to GPSE, can the authors help me understand the difference?
- For SSL the methods are generally pre-trained and then fine-tuned on down-stream tasks, whereas in GPSE the base model GINE is augmented with learned pse and trained from scratch, how does this comparison seem to stand out? The comparison to the baseline of GINE from Hu et al. (2020b) is fair, but how can it be compared to SSL techniques?


[1] GRAPH NEURAL NETWORKS WITH LEARNABLE STRUCTURAL AND POSITIONAL REPRESENTATIONS (ICLR 22)

---

> ### Author Response · Authors · 2023-11-18
> **Reply to Reviewer 6S8q (1/4)**
>
> > The GPSE architecture is unclear in section 2.2...
>
> We thank the reviewer for suggesting improving the clarity of the GPSE’s computation. We have added a more detailed formulation of the GPSE model with equations to Appendix B.1. In brief, GPSE takes input node features as random noise generated from the standard Gaussian distribution, which is then linearly projected to the hidden dimension. It then passes the processed features through a 20-layer GatedGCN with virtual nodes. Finally, the processed hidden representations are projected to predict one of the PSEs via a two-layer MLP.
>
> > The contributions in the paper seem to be limited as many of the methods have been adopted similarly but is unclear how are they different from the existing approach.
>
> We thank the reviewer for raising this question. While we disagree with the statement that “similar methods have already been adopted and that this limits the contributions of the paper”, we understand that this likely stems from the confusion in the scope of GPSE and LPSE papers, which are completely distinct as we have addressed in our global response, “Relation between GPSE and LPSE”. We would nevertheless like to further elucidate our contributions and overall novelty of our method here.
>
> Our novelty lies in the idea that we can **pre-train a positional encoder from random features using a standard MPNN** – a task that is *novel in the literature with no prior attempt* to the best of our knowledge. We not only show that our learned PSEs consistently and significantly outperform conventional PSEs across a wide range of datasets and tasks, but also are much **faster to compute** and have better scaling properties (please see Appendix H for the new computational cost and scaling experiments). Additionally, we show that GPSE is extremely generalizable, a property that is inherently absent in conventional PSEs. As mentioned in the paper, a scalable and generalizable PSE has huge implications for building efficient foundational models for graph data.
>
> Finally, we would like to demonstrate that designing an MPNN that is suitable for this task is also **technically challenging**, even if many submodules we employed are readily available from the literature: As indicated by our ablation studies, typical GNNs (such as GCN or GIN) or network depths (<5 layers) are completely unsuitable for learning PSEs. Instead, we found that gated message passing (GatedGCN) and unusually deep networks are essential for this task. Furthermore, we have (1) discussed these design decisions in Section 2.2, with *theoretical justifications* in Appendix C, (2) showed *evidence supporting the effectiveness* of our choices in Table 1, Figure 2 and Tables F.1 and F.3, and (3) demonstrated the effectiveness of the GPSE model on specific *1-WL toy examples* in appendix E. These extensive discussions we provided and the ablation studies we conducted are not just supplemental information. They stand testament to the effectiveness of GPSE and its ability to capture rich positional and structural representations of graphs in a unified framework for augmenting downstream prediction models.

---

> ### Author Response · Authors · 2023-11-18
> **Reply to Reviewer 6S8q (2/4)**
>
> > Section 2.2 talks more about the general effect of over-smoothing and squashing in graph neural networks, but it is not clear what this has to do with the method of learning effective position encoding for graphs.
>
> We thank the reviewer for raising the question regarding the relevance of over-smoothing/-squashing in our problem setup. We note that these problems are essential to overcome to *effectively learn the positional and structural encodings*, especially for those that require **global views of the graph**. For example, the Laplacian eigenvector corresponding to the first non-trivial eigenvalue, also known as the Fiedler vector, corresponds to the solution of the graph min-max cut problem. Intuitively, this problem requires accessing the global view of the entire graph as it, colloquially, aims to partition the entire graph into two parts with minimal connections.
>
> A straightforward solution to incorporating more global information into the model is by *stacking more message-passing layers* to increase the receptive field and thus effectively expose the model to information beyond the local structure. However, simply stacking more message-passing layers easily leads to the **over-smoothing** problem, where the messages of each node become increasingly uniform as the number of layers increases. Our usage of the gating mechanism, along with residual connection, effectively mitigates this issue while still exposing the model to more non-local information.
>
> Meanwhile, the model may still have difficulty incorporating global information, even after fixing the over-smoothing issue and stacking more layers due to **over-squashing**. Informally, over-squashing can be understood as the difficulty in losslessly sending messages between two nodes across the network. This difficulty is primarily because there are only a few possible routes between the two nodes compared to all other available routes to each of the nodes. We mitigate this problem using a *virtual node* that serves as the global information exchange hub to enable global information exchange, bypassing the “few routes” limitation. Our ablation study shown in Figure 2 showed empirical evidence about the effectiveness of our solutions to over-smothering/-squashing.
>
> To summarize, the over-smoothing and over-squashing problems stem from the requirement of the GPSE model to access global information of the graph effectively, which is necessary for some of the positional encodings we aim to learn. We have also incorporated the above explanations into Appendix C to clarify this relationship, and have adjusted Section 2.2 to refer to this section so that the connection is clearer in the main paper as well.

---

> ### Author Response · Authors · 2023-11-18
> **Reply to Reviewer 6S8q (3/4)**
>
> > Why are GNN models used as a baseline for testing the extreme OOD node-classification benchmarks (table 6) instead it will be interesting to see how transformer+GPSE will perform on the dataset as they don't have the inductive bias of GNNs.
>
> Classical GNNs like GCN and GraphSAGE were used as baselines for the OOD node-classification benchmarks because they are known to outperform graph Transformers and related models (e.g. GPS) on large, homophilic graphs typically used for node classification, for reasons we elaborate on below. Nevertheless, we have run experiments with both graph Transformer and GPS models, we report the results below, the full table is also added to Appendix F, table F.7. In both cases, we see that GPSE leads to clear improvements, even though the overall performance falls short of GCN/SAGE/GATEv2, still leading architectures for large graphs.
>
> Graph Transformers typically have quadratic complexity in the number of nodes to employ global attention between node pairs. These computational requirements become prohibitively large for graphs with nodes in the order of thousands. Additionally, graph Transformers are known to run into over-smoothing issues much more quickly than MPNNs, which leverage the sparsity of these graphs. Especially for homophilic large graphs, the inductive biases of MPNNs are therefore typically preferable to graph Transformers for both theoretical and empirical reasons.
>
> Building scalable graph Transformers is expectedly a major area of interest in graph learning. One simple and effective approach to improve the scalability of both MPNNs and graph Transformers is neighbor sampling, where a subset of neighbors (or nodes in the case of a Transformer) are employed for message-passing. However, we may expect neighbor sampling to simulate the information flow of a graph Transformer only approximately, leading to some loss of information in the process; this is also implied by the comparatively weak performance of the Transformer-based models.
>
> | ****                   | ****      | **arXiv**             | **Proteins**          |
> |------------------------|-----------|--------------------------------|--------------------------------|
> |                        | **+GPSE?** | **ACC %** | **AUROC \%** |
> | GCN                    | NO    | 71.74 ± 0.29                   | 79.91 ± 0.24                   |
> |                        | YES    | 71.67 ± 0.12                   | 79.62 ± 0.12                   |
> | SAGE                   | NO    | 71.74 ± 0.29                   | 80.35 ± 0.07                   |
> |                        | YES    | 72.19 ± 0.32                   | 80.14 ± 0.22                   |
> | GAT(E)v2               | NO    | 71.69 ± 0.21                   | 83.47 ± 0.13                   |
> |                        | YES    | 72.17 ± 0.42                   | 83.51 ± 0.11                   |
> | Transformer            | NO    | 57.00 ± 0.79                   | 73.93 ± 1.44                   |
> |                        | YES    | 59.17 ± 0.21                   | 74.67 ± 0.74                   |
> | GPS                    | NO    | 70.60 ± 0.28                   | 69.55 ± 5.67                   |
> |                        | YES    | 70.89 ± 0.36                   | 72.05 ± 3.75                   |
>
> > The paper [1] has a similar research question and method to GPSE, can the authors help me understand the difference?
>
> We kindly refer the reviewer to our global response, “Relation between GPSE and LPSE,” for a more in-depth explanation of the difference between the two.

---

> ### Author Response · Authors · 2023-11-18
> **Reply to 6S8q (4/4)**
>
> > For SSL the methods are generally pre-trained and then fine-tuned on down-stream tasks, whereas in GPSE the base model GINE is augmented with learned pse and trained from scratch, how does this comparison seem to stand out? The comparison to the baseline of GINE from Hu et al. (2020b) is fair, but how can it be compared to SSL techniques?
>
> We thank the reviewer for the question regarding fine-tuning the GPSE to adapt to the specific downstream task. We have indeed tried this idea on two primary molecular property benchmarking tasks: ZINC and PCQM4Mv2. As we have presented in Appendix Table F.5, fine-tuning could help in certain cases, but there is no consistent pattern showing the superiority of fine-tuning. However, fine-tuning pre-trained models is a vast topic, and how to effectively do so for better downstream performance in our specific setting is an under-explored area. Thus, an exciting future avenue is to study the optimal way to fine-tune GPSE to further adapt to specific downstream datasets.
>
> ---
> We extend our gratitude for the valuable feedback provided by the reviewer. The insights and suggestions have been essential in guiding the improvements to our manuscript. We have carefully addressed each point raised and are prepared for any necessary further discussions or additional experiments. With these revisions, we kindly ask the reviewer to reevaluate their initial assessment and increase their score if deemed appropriate.

---

> ### Author Response · Authors · 2023-11-22
> **Friendly Reminder to Reviewer 6S8q for Open Discussion**
>
> Dear Reviewer 6S8q,
>
> We are immensely grateful for your thorough review and the valuable feedback on our work. In light of your comments, we have meticulously addressed each point in our rebuttal. As the time frame for open discussion is nearing its close, we kindly request your attention to our responses and welcome any further feedback or queries you might have. Your timely and detailed feedback is essential for us to ensure the quality and completeness of our revisions. We greatly appreciate your continued guidance and look forward to your response soon.
>
> Best regards,
>
> The Authors

---

### Author Response · Authors · 2023-11-14
**Reply to reviewer vPPW's anonymity concern**

We thank reviewer vPPW’s attention and concern about the code submission, where there was a potential issue of revealing an author’s identity from the Google Drive link provided to share the pre-trained model weights for improving the reproducibility of our experiments. We apologize for the oversight and remark that revealing our identity using the Google Drive link was not our intention.

That being said, we are confident that **our submission conforms to instructions regarding author anonymity**. The said name is an *alias* of one of the authors, which is not used in publications. Furthermore, no email or affiliation information can be accessed from the provided link. We further point out that this alias information is accessed only by opening Google Drive in the browser and searching for more detailed information proactively (which is not necessary to download the material, as instructions provided in our README). By default, the user is not exposed to any identifier upon following the download instructions or opening the link.

While we agree that it would be best to fully anonymize all links, given the additional steps required to access the alias (which are more convoluted than simply searching for the paper in preprint databases, as reviewers are discouraged from), we respectfully argue that our submission is sufficiently anonymized. We fully understand and appreciate the reviewer’s good intentions, but we would greatly appreciate it if the reviewer could understand the difficulties in fully anonymously hosting data (our pre-trained model weights). We also ask the AC to verify this and would love reviewer vPPW to **score the paper accordingly**, disregarding any anonymity concerns.

---

> ### Comment · Reviewer_vPPW · 2023-11-14
>
> Thank you to the authors for their responses. I feel sorry for any unintended concern, but as a reviewer, I felt obligated to report this issue.
>
> [1] To clarify for AC and the reviewers, when I opened the Google Drive link, I could immediately see the owner's name on the right side of my browser without any additional steps (NOT by searching for more detailed information). So, my discovery of the author's name was merely by clicking the link provided by the authors.
>
>
> [2] Also, by clicking for additional information on the file from the provided link, one can still access the email (AuthorID@msu.edu, which is used in publications) and affiliation information (Michigan State University).
> Additionally, the name I discovered doesn't seem to be a simple alias. It clearly shares the same family name as the author, with a slightly altered given name. I was able to make this connection because the author used the same alias on GitHub, and their linked Twitter account was posting about the content of the paper.
>
>
>
> Unlike the point [1], I leave the judgment of point [2] to AC as it involved an additional step (clicking for information of the file) and further searching (appearing at the top in incognito mode searches). As both a reviewer and a researcher, I express my regret for the necessity of reporting this.
>
>
>
> *I recommend using an anonymous cloud storage service or creating an anonymous account for Google Drive to host data.*

---

### Author Response · Authors · 2023-11-18
**Summary of changes**

We have made several changes to the manuscript, having taken into account valuable feedback from the reviewers. Below is a summary of the changes; the changes in the manuscript are highlighted in blue (except tables and captions) for clarity.

**Updated Experimental Results**
- Added AllPSE (LapPE+ElstaticPE+RWSE+EigValSE+HKdiagSE+CycleSE) results to Tables 2, 3, 4, 5
- Extended MoleculeNet results (Table 4) to include ClinTox, MUV, HIV, with fully extended results in Appendix F, Table F.6
- Extended version of OOD node classification benchmarks (Table 6) to include Transformer and GPS added to Appendix F, Table F.7.

**New Experimental Results**
- As per questions from reviewers regarding computational costs and relative complexity of GPSE, we have conducted two new sets of experiments that measure the computational costs and scaling properties of GPSE and conventional PSEs with respect to (a) increasing dataset size, and (b) increasing graph size. The results are presented in Appendix H.

**Clarifications & Text Improvements**
- Added a discussion of over-smoothing and over-squashing and their relevance to GPSE to Appendix C, Section C.1. Section 2.2 is updated to reflect these changes.
- Improved Appendix E to introduce the WL-test and its implications for GNN expressivity more clearly. Section 3.3 is updated to reflect these changes.

---

### Author Response · Authors · 2023-11-18
**Global Replies (1/3)**

We express our sincere gratitude to the reviewers for their thoughtful comments and feedback. In this global reply, we wish to address and clarify several concerns brought to our attention by multiple reviewers.

## Comprehensive comparison against AllPSE baselines

We thank the reviewers for constructive comments regarding comparing GPSE against the naive baseline to concatenate all PSEs used for training the GPSE model. This method, which we call **AllPSE**, contains LapPE, ElstaticPE, RWSE, HKdiagSE, CycleSE, and EigValsSE. We carried out experiments for AllPSE on all of our primary benchmarking tasks, including the molecular property prediction benchmark (Table 2), ZINC GNN+PSE analysis (Table 3), MoleculeNet benchmark (Table 4), and transferability benchmark (Table 5). In almost all cases (17/18), **GPSE matched or outperformed AllPSE**, indicating the benefit of using a compressed representation of the PSEs rather than simply concatenating them.

We also note that AllPSE leads to worse performance than individual PSEs, e.g., LapPE and RWSE, indicating the challenge of naively using multiple PSEs via concatenation. We hypothesize that this is due to the nature that different PSEs possess vastly different characteristics. Thus, processing them together using a single shared layer failed to preserve the crucial information provided by individual types of PSEs. Meanwhile, GPSE can be seen as a compressed version of multiple PSEs with low-level semantics. This representation contains information about multiple PSEs and encodes them in a shared latent space, making it easier to process them directly in the downstream tasks.

Molecule property prediction benchmark (addition to Table 2):
|		| ZINC (subset) | PCQM4Mv2 (subset) | MolHIV | MolPCBA |
| :----------------- | :---------------: | :---------------: | :----------------: | :---------------: |
|		| MAE ↓	| MAE ↓	| AUROC ↑	| AP ↑		|
|GPS+GPSE	| **0.0648 ± 0.0030** | **0.1196 ± 0.0004** | **0.7815 ± 0.0133** | **0.2911 ± 0.0036** |
|GPS+AllPSE	| 0.0734 ± 0.0030 | 0.1254 ± 0.0011 | 0.7645 ± 0.0236 | 0.2826 ± 0.0001 |
|GPS+(LapPE+RWSE)	| 0.0822 ± 0.0040 | 0.1273 ± 0.0006 | 0.7719 ± 0.0129 | 0.2854 ± 0.0029 |

ZINC GNN+PSE ablation (addition to Table 3):
|		| GCN		| GatedGCN	| GIN		| GINE		| Transformer	|
| :----------------- | :---------------: | :---------------: | :----------------: | :---------------: | :---------------: |
|GPSE	| **0.129 ± 0.003** | **0.113 ± 0.003** | **0.124 ± 0.002** | **0.065 ± 0.003** | **0.189 ± 0.016** |
|AllPSE	| 0.150 ± 0.007 | 0.143 ± 0.007 | 0.153 ± 0.006 | 0.073 ± 0.003 | 0.190 ± 0.008 |
|LapPE+RWSE	| 0.184 ± 0.008 | 0.163 ± 0.009 | 0.174 ± 0.003 | 0.082 ± 0.004 | 0.205 ± 0.007 |

MoleculeNet benchmark (addition to Table 4):
|		| BBBP | BACE | Tox21 | ToxCast | SIDER |
| :----------------- | :---------------: | :---------------: | :----------------: | :---------------: | :---------------: |
|GINE+GPSE	| 66.2 ± 0.9	| **82.72 ± 1.1**	| **77.4 ± 0.8**	| **66.3 ± 0.8**	| 61.1 ± 1.6	|
|GINE+AllPSE	| **67.6 ± 1.2**	| 77.0 ± 4.4		| 75.9 ± 1.0	| 63.9 ± 0.3	| 63.0 ± 0.6	|
|GINE+(LapPE+RWSE)	| 66.2 ± 2.1	| 80.2 ± 3.2		| 75.3 ± 0.7	| 65.1 ± 0.6	| **63.1 ± 0.8**	|

OOD transferability benchmark (addition to Table 5):
|		| Peptides-struct |Peptides-func | CIFAR | MNIST |
| :----------------- | :---------------: | :---------------: | :----------------: | :---------------: |
|		| MAE ↓	| AP ↑		| ACC ↑	| ACC ↑		|
|GPS+GPSE	| **0.2464 ± 0.0025** | **0.6688 ± 0.0151** | **72.31 ± 0.25** | 98.08 ± 0.13 |
|GPS+AllPSE	| 0.2509 ± 0.0028 | 0.6397 ± 0.0092 | 72.05 ± 00.35 | 98.08 ± 00.12 |
|GPS+(LapPE+RWSE)	| 0.2515 ± 0.0020 | 0.6424 ± 0.0093 | 0.7199 ± 0.0013 | 0.9777 ± 0.0009 |

---

> ### Author Response · Authors · 2023-11-18
> **Global Replies (2/3)**
>
> ## Computational complexity of GPSE
>
> Since the backbone of GPSE is a standard MPNN, it has the computation complexity of O(Lm), where L is the number of layers, and m is the number of edges. We highlight that one primary computational benefit of GPSE lies in the fact that, at inference time, its *complexity remains unchanged no matter how many types of PSEs were used* to train the model. In our case, we have six different PSEs, but it is straightforward for future work to include even more complex and specialized PSEs, and would have no effect on the inference time so long as the architecture remains the same.
>
> Another related benefit is that *GPSE could be much faster to compute compared to some PSE that are hard to compute*, such as the CycleSE in our case. Combining the two factors - **constant complexity regardless of the number of and the difficulty to compute the original PSEs** - GPSE demonstrates a strong computational advantage over the traditional approach of computing all individual PSE from scratch. We have also empirically demonstrated the above claims in a newly added set of experiments, where we compare the computation time for different PSEs against GPSE with varying number of graphs and graph sizes. Our results are presented in Appendix [H], where we show that GPSE is not only considerably faster to compute than the explicitly computing PSEs (and orders-of-magnitude faster than computing and combining all PSEs required for a pair comparison), but also scales better than explicit PSE computation. Figure H.1 pertains to the scaling experiments with respect to the number of graphs, while Figure H.2 pertains to the scaling experiments with increasing graph sizes.
>
> Finally, as a concrete example, we note that performing on MolPCBA, computing **AllPSE took about five hours**, while **GPSE only took four minutes** (experiments ran on a compute node with five Intel(R) Xeon(R) Gold 6148 CPUs and a Tesla V100 GPU).
>
> Scaling with respect to the number of graphs:
> | **PSE**        | **%0,1** | **%0,3** | **%0,5** | **%0,8** | **%1** | **%3** | **%5** | **%8** | **%10** | **%25** |
> |----------------|----------|----------|----------|----------|--------|--------|--------|--------|---------|---------|
> | GPSE       | 1s       | 1s       | 1s       | 2s       | 3s     | 9s     | 15s    | 24s    | 32s     | 1m 16s  |
> | AllPSE     | 12s      | 46s      | 41s      | 1m 2s    | 1m 15s | 4m 15s | 7m 13s | 12m 3s | 14m 22s | 39m 52s |
> | LapPE      | 1s       | 1s       | 3s       | 3s       | 5s     | 15s    | 24s    | 35s    | 55s     | 1m 53s  |
> | RWSE       | 1s       | 1s       | 3s       | 3s       | 4s     | 17s    | 25s    | 36s    | 44s     | 1m 41s  |
> | ElstaticPE | 1s       | 1s       | 2s       | 2s       | 3s     | 10s    | 20s    | 33s    | 53s     | 2m 12s  |
> | HKdiagSE   | 1s       | 1s       | 2s       | 3s       | 4s     | 11s    | 22s    | 31s    | 48s     | 3m 14s  |
> | CycleGE    | 6s       | 17s      | 28s      | 44s      | 58s    | 2m 57s | 4m 35s | 7m 12s | 9m 15s  | 27m 20s |
>
> Scaling with respect to graph sizes:
> | **PSE/Graph size**  | **100**  | **300**   | **500**   | **1000** |
> |---------------------|----------|-----------|-----------|----------|
> | GPSE                | 1s       | 7s        | 27s       | 1m 29s   |
> | AllPSE (No CycleGE) | 8s       | 44s       | 2m 15s    | 11m 20s  |
> | LapPE               | 2s       | 9s 250ms  | 34s       | 2m 35s   |
> | RWSE                | 2s       | 9s 760ms  | 31s 480ms | 3m 27s   |
> | ElstaticPE          | 1s 500ms | 10s 670ms | 25s 30ms  | 2m 19s   |
> | HKdiagSE            | 2s       | 13s       | 44s       | 2m 44s   |

---

> ### Author Response · Authors · 2023-11-18
> **Global Replies (3/3)**
>
> ## Relation between GPSE and LPSE
>
> Despite the similarity in the titles between graph positional and structural encoding (GPSE) and learnable positional and structural encoding (LPSE) [1], the two approaches are quite distinct. On the one hand, GPSE aims to *learn to capture diverse PSEs* from the pre-train dataset solely using the graph structure as input. The trained GPSE is then used to extract features from the downstream graphs to augment the downstream model’s performance. On the other hand, LPSE aims to *make better use of input PSEs* by splitting up feature channels and adding an additional positional loss term. We note that **LPSE does not learn to encode PSEs from a pre-training dataset**, hence cannot transfer knowledge between datasets. The main idea of LPSE is twofold. (1) LPSE uses two different hidden feature channels in the message-passing operations, one for the general features and the other for the PSEs. (2) LPSE uses an additional loss term based on the graph Laplacian on the PSE processing channel. We further summarize the difference between the two approaches in three aspects below.
>
> - **Usage of features.** GPSE only uses graph structural information to learn and generate useful PSEs, and the node features are randomly initialized with standard Gaussian. LPSE instead uses *pre-computed* PSE as input to the model (LapPE and RWSE to be more specific).
>
> - **Pre-training.** GPSE needs to be pre-trained on a large-scale graph dataset to learn to encode diverse PSE. LPSE does not require pre-training.
>
> - **Downstream usage.** GPSE extracts positional and structural representations from the graph structures in the downstream dataset. These extracted representations are then used to augment the downstream graph features in place of the standard PSEs. LPSE takes pre-computed PSEs as input instead of outputting PSE representations. Thus, one interesting future avenue to explore is to see whether GPSE can further boost LPSE by feeding GPSE into the model instead of using the default LapPE and RWSE.
>
> [1] Dwivedi, Vijay Prakash, et al. "Graph Neural Networks with Learnable Structural and Positional Representations." International Conference on Learning Representations. 2021.

---

### Meta-Review · Area_Chair_qHyk · 2023-12-09

**Metareview:**

This paper presents GPSE, a graph encoder that learns rich positional and structural encodings (PSEs) for augmenting any GNN. GPSE is highly transferable and can effectively learn a common latent representation for multiple PSEs. The encoder can be used effectively on datasets drawn from significantly different distributions and modalities. The study shows that GPSE-enhanced models can significantly improve performance in certain tasks and perform on par with those that employ explicitly computed PSEs in other cases. GPSE has the potential to be a viable alternative to existing PSE methods and self-supervised pre-training approaches.

Firstly, the studied topic is an important one for graph learning. Secondly, the integration of PSEs within GNNs is relatively novelty for graph representation learning, which may help improve model performance for various tasks. Finally, relatively extensive experiments are provided to validate the transferability of the proposed method.

However, the contribution is not significant enough, since the paper contains too much feature engineering. Secondly, more experiments should be provided to support the method, such as comparison with all the different combination of encodings. Finally, the paper should be polished more carefully, such as adding more details.

**Justification For Why Not Higher Score:**

The contribution is not significant enough, and more experiments should be provided.

**Justification For Why Not Lower Score:**

N/A

---

### Decision · Program_Chairs · 2024-01-16

Reject